# Extended intergenic DNA contributes to neuron-specific expression of neighboring genes in the mammalian nervous system

Ravneet Jaura [1,2,3], Ssu-Yu Yeh [1,2,3], Kaitlin N. Montanera[1,2], Alyssa Ialongo [1,2], Zobia Anwar[1,2], Yiming Lu[2], Kavindu Puwakdandawa[2] & Ho Sung Rhee [1,2 ✉]

Mammalian genomes comprise largely intergenic noncoding DNA with numerous *cis*-regulatory elements. Whether and how the size of intergenic DNA affects gene expression in a tissue-specific manner remain unknown. Here we show that genes with extended intergenic regions are preferentially expressed in neural tissues but repressed in other tissues in mice and humans. Extended intergenic regions contain twice as many active enhancers in neural tissues compared to other tissues. Neural genes with extended intergenic regions are globally co-expressed with neighboring neural genes controlled by distinct enhancers in the shared intergenic regions. Moreover, generic neural genes expressed in multiple tissues have significantly longer intergenic regions than neural genes expressed in fewer tissues. The intergenic regions of the generic neural genes have many tissue-specific active enhancers containing distinct transcription factor binding sites specific to each neural tissue. We also show that genes with extended intergenic regions are enriched for neural genes only in vertebrates. The expansion of intergenic regions may reflect the regulatory complexity of tissue-type-specific gene expression in the nervous system.

[1] Department of Cell & Systems Biology, University of Toronto, Toronto, Ontario M5S3G5, Canada. [2] Department of Biology, University of Toronto, Mississauga, Ontario L5L1C6, Canada. [3] These authors contributed equally: Ravneet Jaura, Ssu-Yu Yeh. ✉email: hosung.rhee@utoronto.ca

More than 98% of the human genome consist of non-coding DNA[1,2]. A large part of noncoding DNA is intergenic regions, located between two neighboring protein-coding genes[3]. Intergenic regions contain a large number of cis-regulatory DNA elements, such as enhancers, which perform a variety of functions leading to gene expression[4,5]. Gene density and the length of intergenic regions are not uniform throughout the genome. The length of intergenic DNA regions has been proposed to be associated with the regulatory complexity of gene expression[6,7]. Previous studies have examined the relationship between the length of intergenic regions and patterns of gene expression in multicellular organisms. In *Caenorhabditis elegans* and *Drosophila melanogaster*, genes with complex functions, such as developmentally regulated genes, are flanked by significantly longer intergenic regions than simple housekeeping genes are[8]. In *Arabidopsis*, longer intergenic regions are positively correlated with higher variability of gene expression levels across tissues[9]. Mammals have hundreds of cell types, that are established by distinct gene expression programs[10]. It is largely unknown whether developmental or tissue-specific genes utilize different amounts of intergenic regulatory elements and which cell-type- or tissue-specific genes have the high regulatory complexity in a diversity of mammalian cell and tissue types.

The regulatory complexity of gene expression involves multiple factors, such as cis-regulatory elements, transcription factors (TFs), and epigenetic marks[11]. Previous studies have revealed that genes with high regulatory complexity, having a high number of cis-regulatory elements, are induced in a cell-type- or stimulus-specific manner during cell lineage commitment in mammals. For example, genes with a higher number of hematopoietic enhancers showed larger changes in expression during hematopoietic differentiation, producing distinct terminally differentiated cell types[12]. Another study found that genes that are induced by immune stimuli have a large set of conserved TF-binding sites that may control specific gene expression programs in the immune system[13]. Multiple studies have identified densely spaced clusters of enhancers, called super-enhancers, in mammalian cell types[14,15]. Super-enhancers comprise a large number of TF-binding sites within a relatively short noncoding DNA region (<12.5 kb) and control the expression of cell identity genes[16,17]. Considering that a short noncoding DNA region may contain a large set of TF-binding sites, it is still not well understood whether genes with high regulatory complexity require long noncoding DNA regions and why genes are differentially spaced throughout the genome.

The intergenic regions in the mammalian genome contain many enhancers, which are often located up to hundreds of kilobase pairs (kb) from the promoter of the nearest gene. Enhancers can activate the expression of their target genes via long-range chromatin interactions[18]. Enhancers contribute to the establishment of developmental gene expression programs that specify and maintain cell identity[19,20]. Chromatin accessibility regulates cell-type- and tissue-specific gene expression programs by allowing TFs to access the enhancer DNA[21]. Developmentally active enhancers are commonly associated with increased chromatin accessibility and enriched for histone H3 lysine 27 acetylation (H3K27ac) in a cell-type- and tissue-specific manner[22]. Long intergenic regions generally have the potential to contain a larger amount of regulatory information than short intergenic regions. We reasoned that genes with longer intergenic DNA might have a greater number of active and accessible enhancers to regulate more complex cellular functions than genes with shorter intergenic DNA. Recent advances in genomic technologies have provided a large number of genome-wide expression and mapping datasets covering hundreds of individual cell and tissue types in mice and humans. Using comprehensive genomic datasets and techniques, we examined whether and why the length of intergenic regions is important for cell- and tissue-type-specific gene expression patterns in mice and humans.

Here, we show that the expression of genes with extended intergenic DNA regions is significantly activated in neural tissues but repressed in non-neural tissues. These genes are associated with neuronal gene annotations and mainly share the extremely long intergenic regions with their neighboring neural genes. These neighboring neural genes are largely co-activated in neural tissues. Co-expression of these neighboring neural genes is mostly controlled by distinct enhancers, rather than common enhancers in their shared intergenic regions. We also demonstrate that neural genes expressed in more cell types have significantly longer intergenic regions than neural genes expressed in fewer cell types. Extended intergenic regions of the neural genes expressed in more cell types have many dispersed active enhancers that contain accessible DNA sites for distinct TFs specific to each neural cell type. Long intergenic length-dependent neural gene enrichment occurs in vertebrates but not in invertebrates. Our results indicate that the amount of cell- and tissue-specific cis-regulatory elements in the extended intergenic regions of neural genes may reflect high levels of regulatory information to accommodate the diverse and complex gene expression patterns in the mammalian nervous system.

## Results

**Long intergenic length-dependent neural gene induction in mice.** To examine the relationship between the length of intergenic regions and the gene induction level in each type of mouse tissue, we used tissue-specific gene expression datasets from previous studies (Supplementary Data 1). We also used the gene expression profile of embryonic stem (ES) cell-derived spinal motor neurons, which can provide a large number of neurons with high purity for genomic mapping and editing analyses[23]. We plotted the intergenic lengths versus the fold-changes in gene expression of each tissue compared to ES cells. The intergenic length of a gene was defined as the sum of the upstream and downstream DNA distances to the ends of the nearest neighboring genes (Supplementary Methods). We used 19,144 annotated protein-coding genes to determine intergenic DNA lengths. We found that genes more highly induced in neural tissues on average had longer intergenic regions than genes highly induced in non-neural tissues (Fig. 1a). The top 5% highly induced genes in neural tissues (919 genes), out of all messenger RNA (mRNA) genes, had approximately 2-fold longer median intergenic lengths than highly induced genes in non-neural tissues (brain: 145 kb, neural tube: 146 kb, motor neurons: 142 kb, non-neural tissues: 54–66 kb, Fig. 1b). We also observed that genes highly induced in neural tissues had long intergenic DNA compared to non-neural tissues including ES cells (Supplementary Fig. 1a). Our results indicated that genes highly induced in neural tissues are associated with long intergenic regions.

Many distinct neural cell types exist in the nervous system[24]. We compared the lengths of intergenic regions of highly induced genes in various types of neurons with those in non-neural tissues and cells in mice (Fig. 1b, Supplementary Fig. 1b). We used specific gene expression datasets for glutamatergic, GABAergic, and cholinergic neurons, as well as other neural tissues in the mouse central nervous system (CNS) and sensory neurons in the peripheral nervous system (PNS)[25–29]. The results revealed that the intergenic lengths for highly induced genes in all neural tissues and cells, examined here, were significantly longer than those in non-neural tissues ($P < 1 \times 10^{-5}$; Fig. 1c). Highly induced genes in each type of neural tissue and neuron had similar lengths of long intergenic DNA (Fig. 1c). We also confirmed that genes

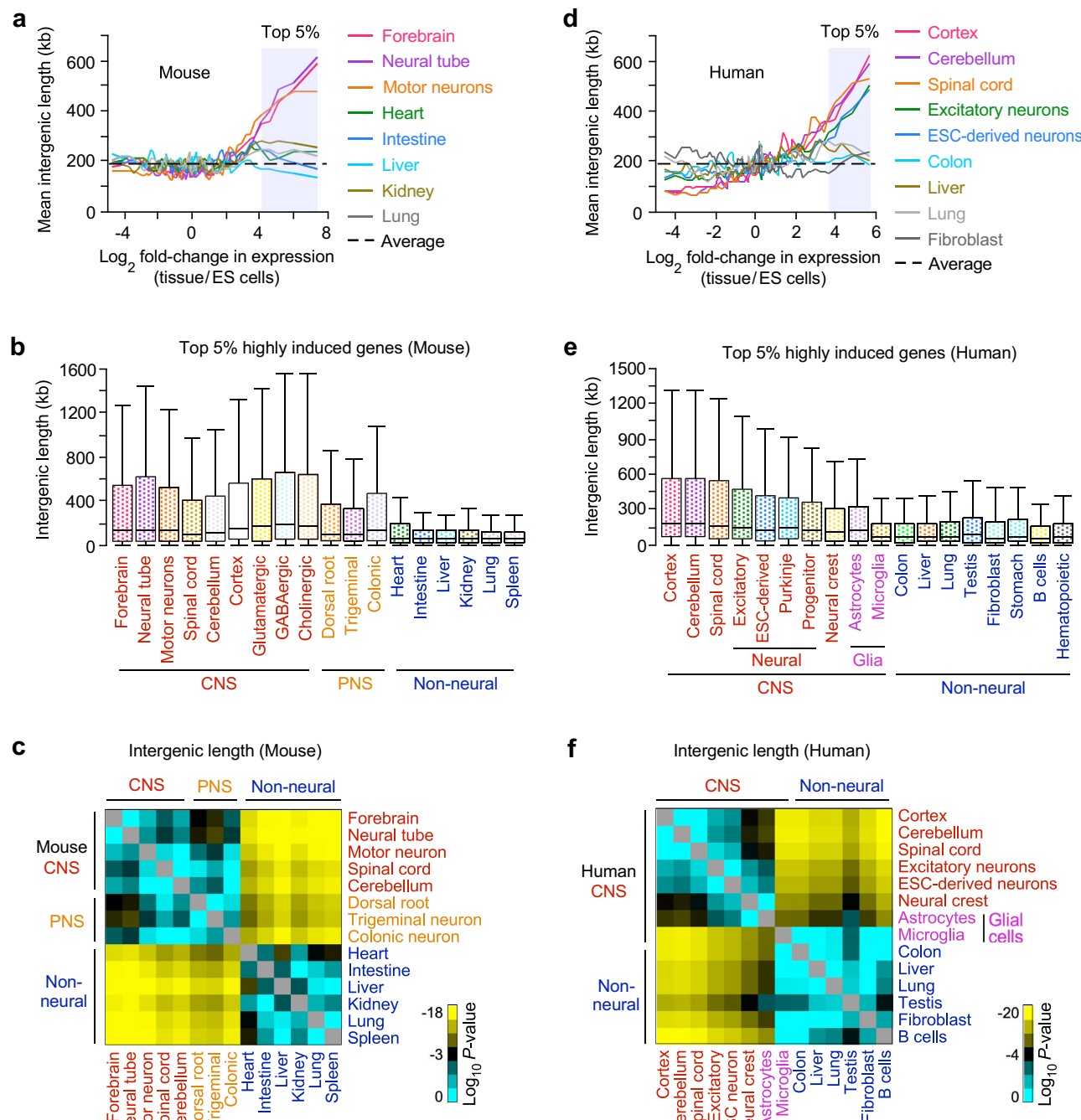

**Fig. 1 Genes highly induced in neurons and neural tissues have significantly extended intergenic DNA. a** Genome-wide relationship between intergenic DNA lengths and gene induction levels, assessed by RNA-seq analysis of postnatal (0 day or P0) mouse tissues (forebrain, heart, intestine, liver, kidney, lung), embryonic (E15.5) neural tube, and ES cell-derived motor neurons (Supplementary Data 1). The lines represent the mean intergenic lengths for genes, binned according to the gene induction level for each tissue compared to ES cells (log$_2$ fold-change in FPKM: fragments per kilobase of transcript per million reads, 200 genes per bin, 18,383 mRNA genes). The dotted line indicates the average intergenic lengths of all the mRNA genes (192.3 kb). **b** Box plots of the intergenic DNA lengths of the top 5% of all the mRNA genes, highly induced in the mouse CNS, PNS, and other tissues (919 genes, Supplementary Data 2), shown in the shaded area in (**a**). Dorsal root, trigeminal, and colonic neurons represent sensory neurons in the PNS. The box plots show the median (line in a box), first-to-third quartiles (boxes), and 1.5 × the interquartile range (whiskers). **c** Heatmap of the *P*-value (two-sided, two-sample *t*-test) for the difference in intergenic lengths of the top 5% of all the mRNA genes, highly induced in each tissue (919 genes) between pairwise combinations of the representative tissues shown in (**b**) (Supplementary Data 3). *P*-values were not adjusted for multiple comparisons. Yellow represents significant *P*-values, and black and cyan represent less or non-significant *P*-values. **d** Same analyses as shown in (**a**) except human tissues and cells. The lines represent the mean intergenic lengths for genes, binned according to the gene induction level for each tissue compared to ES cells (200 genes per bin, 19,268 mRNA genes, Supplementary Data 4). The dotted line indicates the average intergenic lengths of all the mRNA genes (197.3 kb). **e**, **f** Same analyses as shown in (**b**, **c**) except human tissues and cells, reported in Supplementary Data 1. The top 5% highly induced genes in each tissue and cell type (963 genes), out of all the mRNA genes, were used for the analyses.

highly induced in mouse neural tissues had preferentially longer intergenic regions in the embryonic (E14.5), postnatal (P0), and adult stages than those in non-neural tissues (Fig. 1a, Supplementary Fig. 1c, d). This analysis suggests that the expression of neural genes might be activated by long intergenic DNA from the embryonic stage at which neural cell identity is acquired.

**Genes linked to human neurons and diseases have long intergenic regions.** Humans are complex organisms with more diverse and highly specialized cell and tissue types compared to other organisms[10]. We tested whether genes highly induced in diverse tissues and cells in the human nervous system have longer intergenic regions than genes highly induced in non-neural tissues. We examined the relationship between intergenic lengths and gene induction levels across 26 cell and tissue types, using published RNA-seq datasets (Supplementary Data 1). In agreement with our results using mouse tissues, we revealed that the top 5% highly induced genes in neural tissues and neurons (963 genes), out of all the mRNA genes, had markedly longer intergenic regions than those in other human tissues (Fig. 1d–f, Supplementary Fig. 2a, b). Thus, our findings indicate that long intergenic regions are significantly associated with gene expression in neurons and neural tissues across the diverse human cell and tissue types.

Genetic variations in noncoding regulatory DNA are often linked to genetic risk for neurological diseases and disorders such as Parkinson's disease (PD), which is characterized by the degeneration of excitatory neurons in the brain[30,31]. Recently, a high number of single nucleotide polymorphisms (SNPs) associated with PD genetic risk have been identified in noncoding DNA regions. We investigated whether genes containing SNPs in their intergenic regions had long intergenic DNA by using 9064 SNPs, identified from genome-wide association studies of PD patients[30]. Our result showed that a total of 49.2% of SNPs associated with PD risk (4460 out of 9064) resided within the intergenic regions of 684 genes. Gene ontology analysis showed that these genes were enriched for neural gene annotations (Supplementary Fig. 2c). We compared the intergenic lengths of these 684 genes with the intergenic lengths of highly induced genes in the human brain and non-neural tissues. We found that genes containing these intergenic SNPs associated with PD risk had long intergenic DNA (median 117 kb), with lengths not significantly different from the intergenic lengths of the highly induced genes in the cortex and excitatory neurons (935 genes, $P > 0.02$; Supplementary Fig. 2d). These intergenic SNP-containing genes had significantly longer intergenic regions than highly induced genes in non-neural tissues ($P < 1 \times 10^{-10}$). We demonstrated that genes associated with PD risk contain a large number of SNPs in their long intergenic DNA. This observation suggests that *cis*-regulatory elements in long intergenic regions are involved in neural functions, whose misregulation is associated with neurodegeneration in neurological disorders.

**Highly induced genes in non-neural glial cells have long intergenic regions.** Nervous tissue consists of multiple cell types. For example, brain tissue contains not only neurons but also non-neural cells called glial cells, which provide support and immune protection for neural cells[32]. We asked the question of whether the correlation between intergenic lengths and gene induction levels is specific for neurons or neural tissues containing neurons and glial cells. To investigate this question, we analyzed gene expression datasets of the distinct non-neural glial cells, isolated by fluorescence-activated cell sorting (FACS), or identified by single-cell RNA-seq[33–35] (Supplementary Data 1). Our analysis included various non-neural glial cells, such as astrocytes (Aldh1l1+, Acsa2+), oligodendrocytes, microglial cells (Cd11b+, Cd45+), and myeloid

cells in the mouse brain. We found that highly induced genes in astrocytes and oligodendrocytes, which support axons and synaptic functions, contained significantly long intergenic DNA than non-neural tissues, as was observed in neurons ($P < 1 \times 10^{-11}$; Fig. 2a, b). However, there was no relationship between intergenic lengths and gene induction levels in other glial cells such as microglia and myeloid cells, which function as immune and blood cells in the brain, respectively. In the human brain, we also confirmed that highly induced genes in astrocytes, but not in microglial cells, had significantly longer intergenic regions than non-neural tissues ($P < 1 \times 10^{-16}$; Fig. 1e, f, Supplementary Fig. 2b). Our results indicated that intergenic length-dependent gene expression is observed in non-neural cells such as astrocytes and oligodendrocytes, which support axons and maintain synaptic functions in the mammalian CNS.

Glial cells, such as astrocytes and oligodendrocytes, express many neural genes, such as synapse-regulating genes and neurotransmitter receptor genes, as well as glial-cell-specific genes in the CNS[36–38]. We asked whether glial-cell-specific genes had long intergenic regions and whether the neural genes commonly expressed in neurons and glial cells had long intergenic regions. After eliminating the housekeeping genes (755 genes), we selected the top 10% highly induced genes out of all the induced mRNA genes (1596 out of 15,964) in mouse glutamatergic neurons, astrocytes, and oligodendrocytes (Supplementary Methods). Then, we compared the length of the intergenic regions in the genes that overlapped among the 3 CNS cell types. We found that 31.8% of the genes highly induced in each cell type (508 out of 1596) were common in all 3 cell types (Fig. 2c, d) and had significantly longer intergenic regions than the genes expressed in only 1 or 2 CNS cell types ($P < 1 \times 10^{-5}$; Fig. 2e). The highly induced genes (508 genes) in all 3 cell types were enriched with neural gene ontologies, such as nervous system development and synaptic regulation (Supplementary Fig. 3a). On the other hand, 575 and 335 genes were induced only in 1 cell type, astrocytes or oligodendrocytes, respectively (Fig. 2c); they were enriched with the glial cell gene ontologies, such as metabolic support and immune function for neurons (Supplementary Fig. 3b). The glial-cell-specific genes also had significantly longer intergenic regions than the genes (13,817 genes) not expressed in the 3 cell types (astrocyte-specific genes: median 72.7 kb, oligodendrocyte-specific genes: median 69.2 kb, other genes: median 37.3–49.5, Fig. 2e), indicating that glial-cell-specific genes have long intergenic regions. These results suggest that both neural and glial-cell-specific genes may utilize their long intergenic regions to express the neural and glial-cell-specific genes in the mammalian CNS.

**Genes with long intergenic DNA are repressed in non-neural tissues.** We next investigated whether genes with extremely long intergenic DNA regions (>500 kb, termed EID genes) were strongly expressed in neural tissues compared with non-neural tissues at the genomic level. We defined EID genes as genes flanked by more than 500 kb of intergenic DNA (1621 genes; Fig. 3a, Supplementary Data 1). EID genes were highly induced in tissues of the forebrain, neural tube, and motor neurons, compared to non-neural tissues including ES cells (Log$_2$ fold-change in mRNA expression, Supplementary Fig. 4a, b). We further considered whether neural tissue-specific induction of EID genes might be attributable to the high expression of EID genes in neural tissues or to the repression of EID genes in non-neural tissues. To answer this question, we compared the absolute gene expression levels (Log$_{10}$ mRNA expression) in each tissue with the intergenic DNA lengths. Unexpectedly, we found that EID genes were moderately, rather than highly, expressed in neural

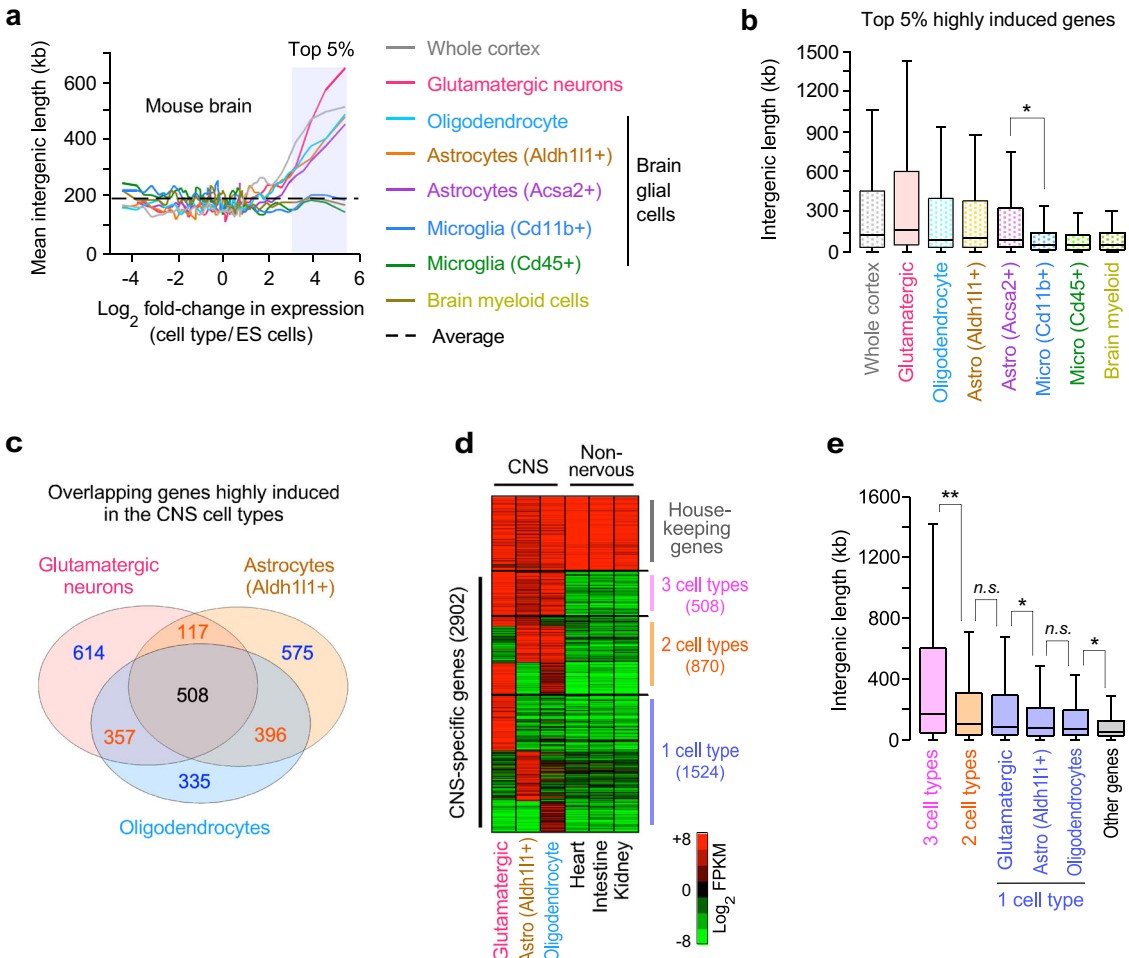

**Fig. 2 Highly induced genes in neural supporting glial cells have extended intergenic lengths. a** Genome-wide relationship between intergenic DNA lengths and gene induction levels in neurons and various glial cell types in the mouse brain. The lines represent the mean intergenic lengths for the genes, binned according to the gene induction level for each cell type compared to ES cells (log$_2$ fold-change in FPKM; 200 genes per bin; 18,383 mRNA genes; Supplementary Data 2). **b** Box plots of the intergenic DNA lengths of the top 5% highly induced genes in each cell type, out of all the mRNA genes (919 out of 18,383), shown in the shaded area in (**a**). The box plots indicate the median (line in a box), first-to-third quartiles (boxes), and 1.5 × the interquartile range (whiskers). *$P = 1.05 \times 10^{-10}$; two-sided, two-sample $t$-test (Supplementary Data 3). **c** The Venn diagram of the overlap of highly induced genes (2902 in total) between 3 CNS cell types (glutamatergic neurons, astrocytes, and oligodendrocytes) in mice. The top 10% highly induced genes in each cell type, out of all the mRNA genes, were used. 508 genes were induced in all 3 CNS cell types (Supplementary Methods). **d** Expression levels of 755 housekeeping (HK) genes and 2902 genes, based on the Venn diagram analysis in (**c**), in the 3 CNS cell types and 3 non-neural tissues (the mouse 8-week heart, intestine, and kidney). The HK genes were induced >2-fold compared to ES cells in all 6 cell types. 870 genes were induced in 2 CNS cell types and 1524 genes were induced in only 1 CNS cell type. The FPKM values were median normalized and log$_2$ transformed (Supplementary Data 5). **e** Box plots of the intergenic DNA lengths of the groups of genes (2902 genes) defined in (**c**, **d**) and all other genes (13,817 genes). The box plots show the median (line in a box), first-to-third quartiles (boxes), and 1.5 × the interquartile range (whiskers). **$P = 3.93 \times 10^{-6}$; *$P < 3.70 \times 10^{-3}$; *n.s.*, non-significant $P > 0.44$; two-sided, two-sample $t$-test.

tissues (Fig. 3b). For example, EID genes were highly induced in the forebrain because of the preferential repression of EID genes in non-neural tissues. Our results can be supported by the finding that cell-type-specific genes are repressed in many other cell types until a particular cellular identity or process requires gene activation[39–41]. We also observed that genes with short intergenic regions were strongly expressed in all neural and non-neural tissues and cells examined here (Fig. 3b). These results suggest that EID genes might be associated with neural functions, whereas genes with short intergenic regions might be associated with housekeeping or simple cellular functions. Indeed, gene ontology analysis confirmed that EID genes were significantly enriched in neural gene annotations, whereas genes with short intergenic regions were enriched in basic cellular processes (Fig. 3c).

**Long intergenic regions have more enhancers in neurons than other cell types**. We then asked how EID genes are associated with patterns of neural gene expression. The nervous system is the most complex system in mammals[42]. The diversity and complexity of neural cell types are reflected in the number of TFs that regulate complex gene expression programs in the nervous system[43,44]. Specific gene expression in the nervous system can be achieved by the binding of a set of distinct TFs to accessible DNA regions in a neural cell-type- and cell-stage-specific manner. We hypothesized that EID genes in neural tissues might require a larger number of accessible regions in their long intergenic regions than those in non-neural tissues, in order to regulate the diverse gene expression patterns in the nervous system. To test this hypothesis, we compared the number of accessible regions per intergenic region with the intergenic lengths by using assays

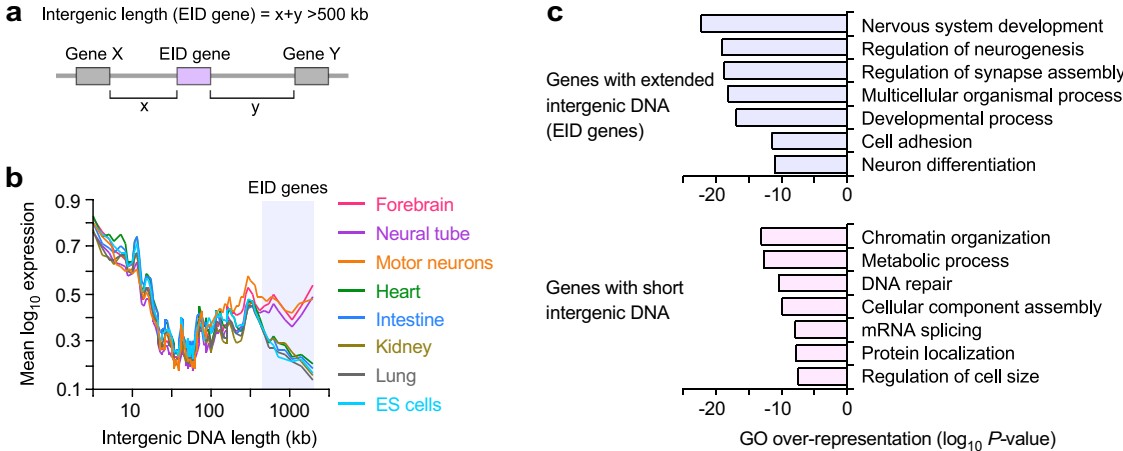

**Fig. 3 Genes with long intergenic DNA are not highly expressed in neural tissues but repressed in non-neural tissues. a** Illustration of intergenic lengths for EID genes (genes with extremely long intergenic DNA). EID genes have longer than 500 kb of intergenic DNA lengths. "x" and "y" indicate the upstream and downstream DNA distances to the ends of the nearest neighboring genes, X and Y, from the ends of focal genes. **b** Absolute gene expression levels ($\log_{10}$ FPKM) in tissues shown in Fig. 1a and mouse ES cells, depending on the intergenic DNA lengths. The lines indicate the mean gene expression level for genes, binned according to the intergenic DNA lengths for each tissue (200 genes per bin, 18,383 mRNA genes). EID genes (shaded) are repressed in non-neural tissues and ES cells. **c** Gene ontology (GO) analysis for EID genes (1525 genes), shown in the shaded area in (**b**), and genes having short intergenic DNA regions (1848 genes with <10 kb of intergenic DNA lengths). The most enriched seven non-redundant GO terms are shown (Supplementary Data 6, 7). A one-sided Fisher's exact test was used to calculate P-values in gene set enrichment analysis. P-values were not adjusted for multiple comparisons.

for transposase-accessible chromatin with sequencing (ATAC-seq) datasets in various tissues (Supplementary Data 1). Remarkably, we found that EID genes in neural tissues as a population contain twice as many accessible regions in their intergenic regions as other tissues (Fig. 4a, b). The number of accessible regions per intergenic region in each type of neural tissue was significantly higher than in non-neural tissues ($P < 1 \times 10^{-12}$; Fig. 4c).

Multiple cell-type-specific enhancers are necessary to activate the diverse expression patterns of a neural gene with spatial and temporal resolution in the nervous system[44,45]. We hypothesized that EID genes in neural tissues have more active enhancers than those in other tissues, reflecting the diversity of neural cell types. Indeed, we found that a long intergenic region globally contained a greater number of H3K27ac-enriched active enhancers in neural tissues than in other tissues, as determined by chromatin immunoprecipitation coupled with sequencing (ChIP-seq) (Fig. 4d, e). More specifically, EID genes had significantly more H3K27ac-enriched peaks per intergenic region in neural tissues than in non-neural tissues ($P < 1 \times 10^{-7}$; Fig. 4f). In contrast, the neural-tissue-specific H3K27ac peaks were not enriched in the intergenic regions of EID genes in non-neural tissues, indicating that neuronal enhancers were repressed in non-neural tissues. Our findings suggest that a higher number of intergenic accessible DNA regions may be available for TFs to bind to enhancers of EID genes in neural tissues than in other tissues. This allows more active enhancers to be established, giving rise to diverse neural gene expression programs.

As observed in super-enhancers in several cell types and tissues[14,15], a relatively short noncoding DNA region (<12.5 kb) around cell identity genes may contain many densely spaced enhancers. We examined whether enhancers of highly induced genes in each tissue were dispersed or clustered in their intergenic regions. We found that the distances between the nearest active enhancers within an intergenic region of a highly induced gene in neural tissues were significantly longer than those in other tissues ($P < 1 \times 10^{-10}$; Supplementary Fig. 5). Neural tissues had more than 2-fold longer median distances between the closest

intergenic enhancers per gene than non-neural tissues (neural: 13.6–16.8 kb, non-neural: 4.1–6.3 kb). This result indicated that enhancers associated with neural genes are more likely to be dispersed throughout long intergenic regions, whereas enhancers associated with non-neural genes are often concentrated within relatively short intergenic regions, as observed in super-enhancers in many non-neural cell types[14,15]. We establish that a large number of dispersed enhancers are uniquely organized in long intergenic regions of neural genes. Taken together, our results suggest that an increased number of enhancers, dispersed in the long intergenic regions of neural genes, help them exploit high levels of regulatory information to accommodate the diverse and complex gene expression patterns found in the nervous system.

**Extended intergenic regions have multiple tissue-specific neural TF-binding sites.** Next, we asked whether the long intergenic regions contained common TF-binding sites in multiple neural tissues or had distinct TF-binding sites in each neural tissue. We defined generic neural genes as the genes highly expressed in 4 neural tissues examined here (the postnatal P0 forebrain, adult cortex, adult hippocampus, and embryonic motor neurons) but not expressed in non-neural tissues (the adult heart, intestine, kidney, and lung) in mice (Fig. 5a). First, we selected 625 generic neural genes with extended intergenic regions (intergenic lengths >100 kb). Then, we identified the commonly and differentially accessible DNA sites in the intergenic regions of the generic neural genes using available ATAC-seq datasets (Supplementary Data 1). Subsequently, we identified 15,401 ATAC-seq peaks in the intergenic regions of the 625 generic neural genes in the 4 neural tissues. More than 71% of the intergenic ATAC-seq peaks (11,004 out of 15,401) were distinct in each neural tissue, whereas only 3.4% of the intergenic ATAC-seq peaks (521 out of 15,401) were common in all 4 neural tissues (Fig. 5b, Supplementary Fig. 6a). Moreover, 496 out of the 625 intergenic regions of the generic neural genes with >100 kb intergenic lengths had distinct tissue-specific ATAC-seq peaks in 3 or 4 neural tissues (Fig. 5c, Supplementary Fig. 6b).

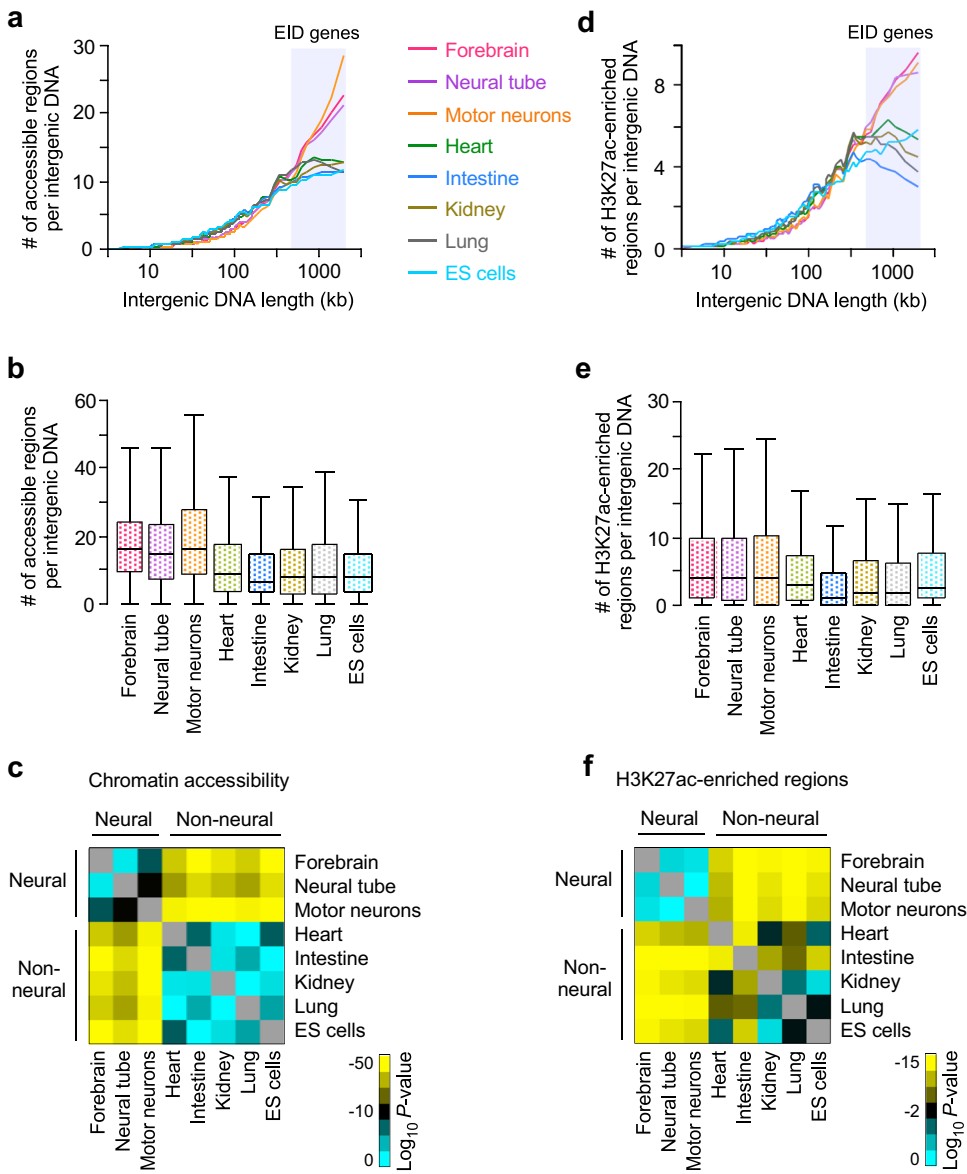

**Fig. 4 Genes with long intergenic DNA have significantly more accessible and active enhancers in neural tissues than other tissues. a, d** Average number of ATAC-seq peaks (**a**) and ChIP-seq H3K27ac peaks (**d**) per intergenic DNA region for genes, depending on the intergenic DNA lengths, in mouse tissues and cells shown in Fig. 3b. The lines indicate the average number of peaks for genes, binned according to the intergenic DNA lengths for each tissue and cell type (200 genes per bin, 18,383 mRNA genes, Supplementary Data 8). **b, e** Box plots of the number of ATAC-seq peaks (**b**) and ChIP-seq H3K27ac-enriched peaks (**e**) per intergenic DNA region of EID genes in each tissue and cell type (1621 genes), shown in the shaded areas in (**a, d**). The box plots show the median (line in a box), first-to-third quartiles (box), and 1.5 × the interquartile range (whiskers). **c, f** Heatmap of the $P$-values (two-sided, two-sample $t$-test) for the difference in the number of ATAC-seq peaks (**c**) and H3K27ac-enriched peaks (**f**) per intergenic DNA regions of EID genes (1621 genes) between all pairwise combinations of tissues and cells, shown in (**b, e**). $P$-values were not adjusted for multiple comparisons. Yellow represents significant $P$-values, and black and cyan represent less or non-significant $P$-values (Supplementary Data 9).

Furthermore, the generic neural genes had an average of 8 tissue-specific ATAC-seq peaks per intergenic region. Our findings suggest that most intergenic accessible DNA sites of the generic neural genes are differentially distributed across neural tissues.

Considering the binding of tissue-specific TFs to accessible DNA sites, we next investigated which TFs potentially bound to the accessible DNA sites in the extended intergenic regions in the 4 neural tissues. Using the de novo motif discovery[46], we found that the tissue-specific ATAC-seq peaks of the generic neural genes were enriched with the distinct DNA sequence motifs in each neural tissue (Fig. 5b). Surprisingly, the most significantly enriched DNA motifs, found in each neural tissue, preferentially occurred in the ATAC-seq peaks specific in one tissue, not in the

other tissues (Fig. 5d). These tissue-specific DNA motifs included the TF-binding sequences for neural TFs, such as RFX in the forebrain, AP-1 in the cortex, NeuroD in the hippocampus, and Onecut in the motor neurons[47–51]. We also found a DNA binding motif for the basic helix loop helix (bHLH) TF Ascl1 (CCAGCTG), which had 1 nucleotide difference from the bHLH TF NeuroD motif (CCATCTG) (Supplementary Fig. 7a). The Ascl1 motif was found in the accessible DNA sites in the 4 neural tissues rather than in a specific neural tissue (Supplementary Fig. 7b). These results suggest that the proneural TF, Ascl1, may be involved in neural gene expression in many tissues, while NeuroD TFs may be associated with the hippocampus-specific gene expression. In addition, the commonly accessible DNA sites

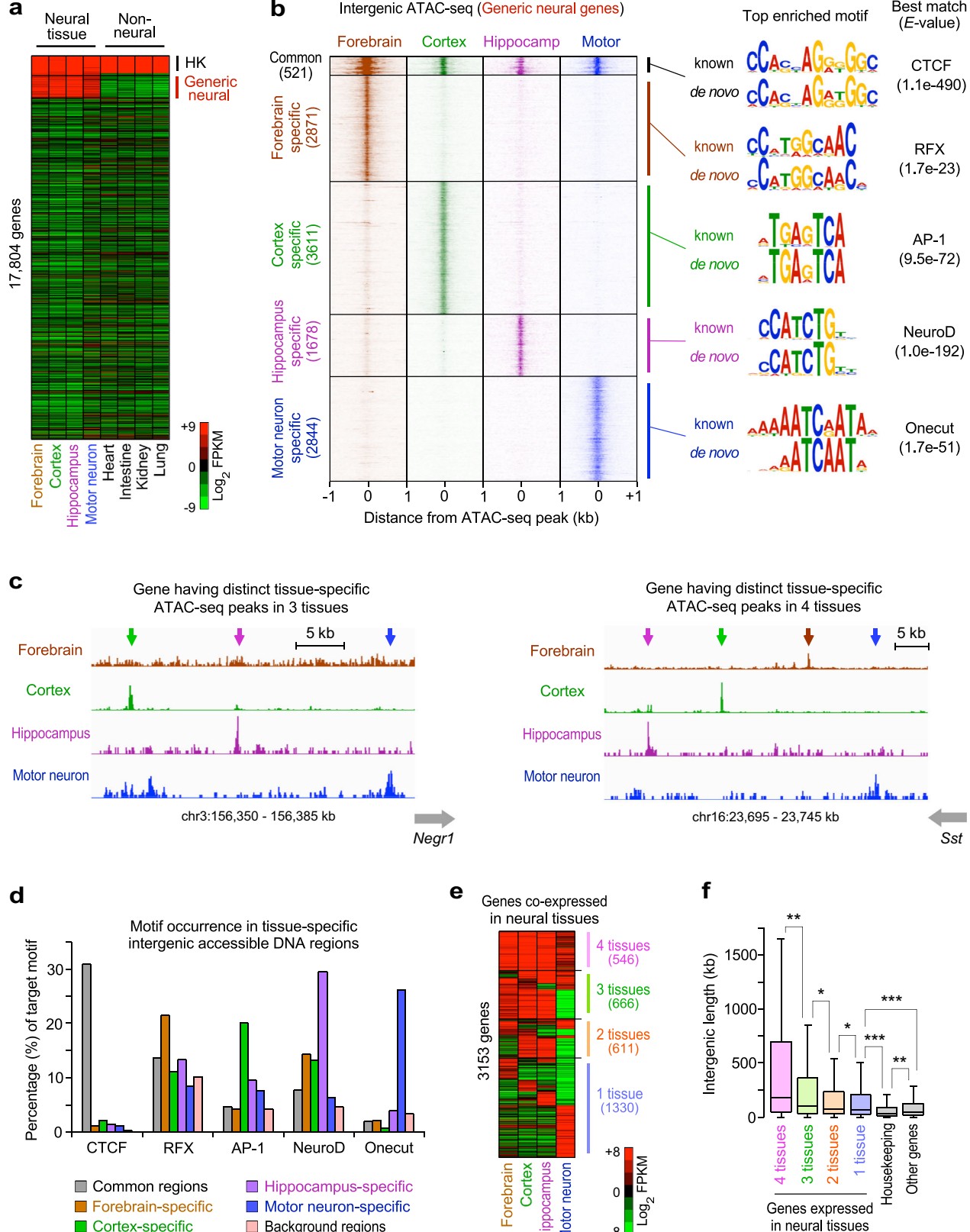

(521 peaks) were preferentially enriched with a DNA binding motif for the insulator binding protein, CTCF (Fig. 5b, d). Using ChIP-seq analyses, we confirmed that the commonly accessible DNA sites were mostly bound by CTCF in all 4 neural tissues and ES cells (Supplementary Fig. 8a). Our results suggest that the commonly accessible DNA sites bound by CTCF may be associated with an insulator or a topologically associated domain (TAD) boundary in the extended intergenic regions in neural and non-neural tissues[52,53]. We also found that the ATAC-seq peaks specific in each tissue were enriched with H3K27ac in a tissue-specific manner (Supplementary Fig. 8b). Together, our results indicate that the generic neural genes have a large number of

**Fig. 5 Extended intergenic regions of generic neural genes have distinct enhancers containing tissue-specific TF-binding sites. a** Expression levels of 17,804 all mRNA genes, expressed in 8 tissues. They include the housekeeping (HK; 790 genes), which were induced >2-fold in all 8 mouse tissues, and generic neural genes (1131 genes), which were induced >2-fold only in 4 neural tissues. The FPKM values were median normalized and $\log_2$ transformed. **b** ATAC-seq mapping in the intergenic accessible DNA sites of the 625 generic neural genes with >100 kb intergenic regions in 4 neural tissues, grouped by commonly accessible DNA sites (521 peaks) and 4 groups of the tissue-specific accessible DNA sites (11,004 peaks). The de novo DNA motifs represent the most (top) enriched motif within 50 bp from the midpoints of ATAC-seq peaks in each group (Supplementary Data 10). **c** Examples of intergenic regions with multiple tissue-specific ATAC-seq peaks (arrows) in multiple neural tissues. The indicated loci show the intergenic regions of the neural genes, *Negr1* (Neuronal growth regulator) and *Sst* (Somatostatin), with distinct tissue-specific ATAC-peaks. **d** Percentage of the occurrence of the DNA motifs within 100 bp from the midpoints of the tissue-specific intergenic ATAC-seq peaks, identified in (**b**) and background genomic regions. All MEME motif format files, used here, were reported in Supplementary Data 11. **e** Expression levels of 3153 mRNA genes grouped by the number of neural tissues expressing the same set of neural genes and tissue-specific genes. The top 10% highly induced genes in each tissue, out of all the mRNA genes (1780 out of 17,804), were used to define the overlap of induced genes between each tissue (Supplementary Methods). The housekeeping genes were not included as the overlapped genes. The FPKM values were median normalized and $\log_2$ transformed. **f** Box plots of the intergenic DNA lengths of the 4 groups of neural genes grouped in (**e**), 790 housekeeping genes defined in (**a**), and all other genes (13,861 genes). The box plots show the median (line in a box), first-to-third quartiles (boxes), and 1.5 × the interquartile range (whiskers). ***$P < 2.73 \times 10^{-13}$; **$P < 9.35 \times 10^{-4}$; *$P < 4.12 \times 10^{-2}$; two-sided, two-sample $t$-test.

---

*cis*-regulatory elements for the distinct neural TF binding in their extended intergenic regions. As a result, various neural TFs may be involved in the activation of tissue- and developmental-specific intergenic enhancers with spatial and temporal tissue specificity.

Because the intergenic regions of the generic neural genes had many dispersed enhancers containing binding sites for tissue-specific TFs (Supplementary Fig. 5), we further hypothesized that the neural genes expressed in more tissues should have longer intergenic regions containing more tissue-specific TF-binding sites than the neural genes expressed in fewer tissues. Thus, we evaluated this possibility by comparing the intergenic lengths of the neural genes commonly expressed in the 4 neural tissues (the P0 forebrain, adult cortex, adult hippocampus, and embryonic motor neurons) with the intergenic lengths of the genes expressed in fewer neural tissue (Fig. 5e). Interestingly, we uncovered a positive relationship between the number of neural tissues expressing the same set of neural genes and their intergenic DNA lengths (Fig. 5f). We found that the genes expressed in more neural tissues had significantly longer intergenic regions than the genes expressed in fewer neural tissues. The genes expressed in only 1 neural tissue had shorter intergenic lengths than the genes expressed in multiple neural tissues but had significantly longer intergenic lengths than non-neural genes ($P < 1 \times 10^{-3}$; Fig. 5f). The genes expressed in 1 neural tissue were also preferentially enriched with the neural genes annotated in biological processes such as neural development and neurogenesis (Supplementary Fig. 8c). In addition, the genes expressed in 1 neural tissue had fewer accessible DNA sites in their intergenic regions than the genes expressed in more neural tissues (Supplementary Fig. 8d), suggesting that tissue-specific neural genes may utilize fewer enhancers than the generic neural genes. Furthermore, we observed that the neural genes commonly expressed in glutamatergic neurons, astrocytes, and oligodendrocytes had significantly longer intergenic lengths than the genes expressed in 1 or 2 cell types in the mouse CNS ($P < 1 \times 10^{-5}$; Fig. 2e).

We next tested whether constitutive genes required for basic cellular functions, such as housekeeping genes, also had long intergenic regions like the generic neural genes. We selected 790 housekeeping genes highly induced in all the 4 neural and 4 non-neural tissues (Fig. 5a). We found that the housekeeping genes had significantly shorter intergenic lengths than other genes, such as tissue-specific neural genes ($P < 1 \times 10^{-3}$; Fig. 5f). Our findings indicated that the correlation between the number of tissues expressing the same set of genes and the intergenic DNA lengths of the genes occurred in neural genes, not in housekeeping genes. In summary, our results suggest that the extended intergenic regions of neural genes contain the distinct tissue-specific neural

TF-binding sites that are necessary for diverse gene expression patterns in the complex mammalian nervous system.

**Co-expression of neighboring neural genes with extended intergenic regions.** The change in the gene expression of a focal gene may be correlated with the changes in the expression of its neighboring genes in humans[54,55]. The co-expression of neighboring genes has been observed in constitutively active genes, such as housekeeping genes[56,57]. Our finding that genes with extended intergenic regions were associated with neural gene expression raised the questions of whether extended intergenic regions were more likely to be flanked by two neural genes and whether neighboring neural genes sharing extended intergenic regions were co-expressed at the genome-wide level. Because highly induced genes in neural tissues preferentially have extended intergenic regions, we hypothesized that two neighboring genes that share extended intergenic regions might be highly co-expressed in neural tissues.

To investigate this possibility, we plotted the genome-wide organization of the transcription start sites (TSSs; orange in Fig. 6a) and transcription end sites (TESs; cyan) of all the protein-coding genes in mice (18,040 genes), and sorted them by the lengths of their shared intergenic DNA. We classified all the genes into 4 groups based on the orientation of neighboring gene pairs (Fig. 6a): head-to-head (H-H) genes where the two genes were located on the opposite DNA strands and transcribed divergently; tail-to-head (T-H) genes where the two genes were located on the upper DNA strand and transcribed in the same direction; head-to-tail (H-T) genes where the two genes were located on the lower DNA strand and transcribed in the same direction; and tail-to-tail (T-T) genes where the two genes are located on the opposite DNA strands and transcribed convergently. We defined the two neighboring genes as the left and right genes located at the earlier and later genomic coordinates of the shared intergenic regions, respectively. Then, we examined the tissue-specific gene induction levels of the left and right genes in neural and non-neural tissues in the 4 groups, depending on their shared intergenic DNA lengths. We found that both left and right genes sharing extended intergenic regions were highly induced to a similar level in the neural tissues, including the mouse forebrain, neural tube, glutamatergic, and cholinergic neurons, but not in non-neural tissues (Fig. 6b). Interestingly, the 4 groups of the neighboring gene pairs behaved similarly (Fig. 6c), indicating a positive relationship between intergenic lengths and gene induction levels. The genes with >500 kb shared intergenic regions showed significantly higher gene induction levels in neural tissues than

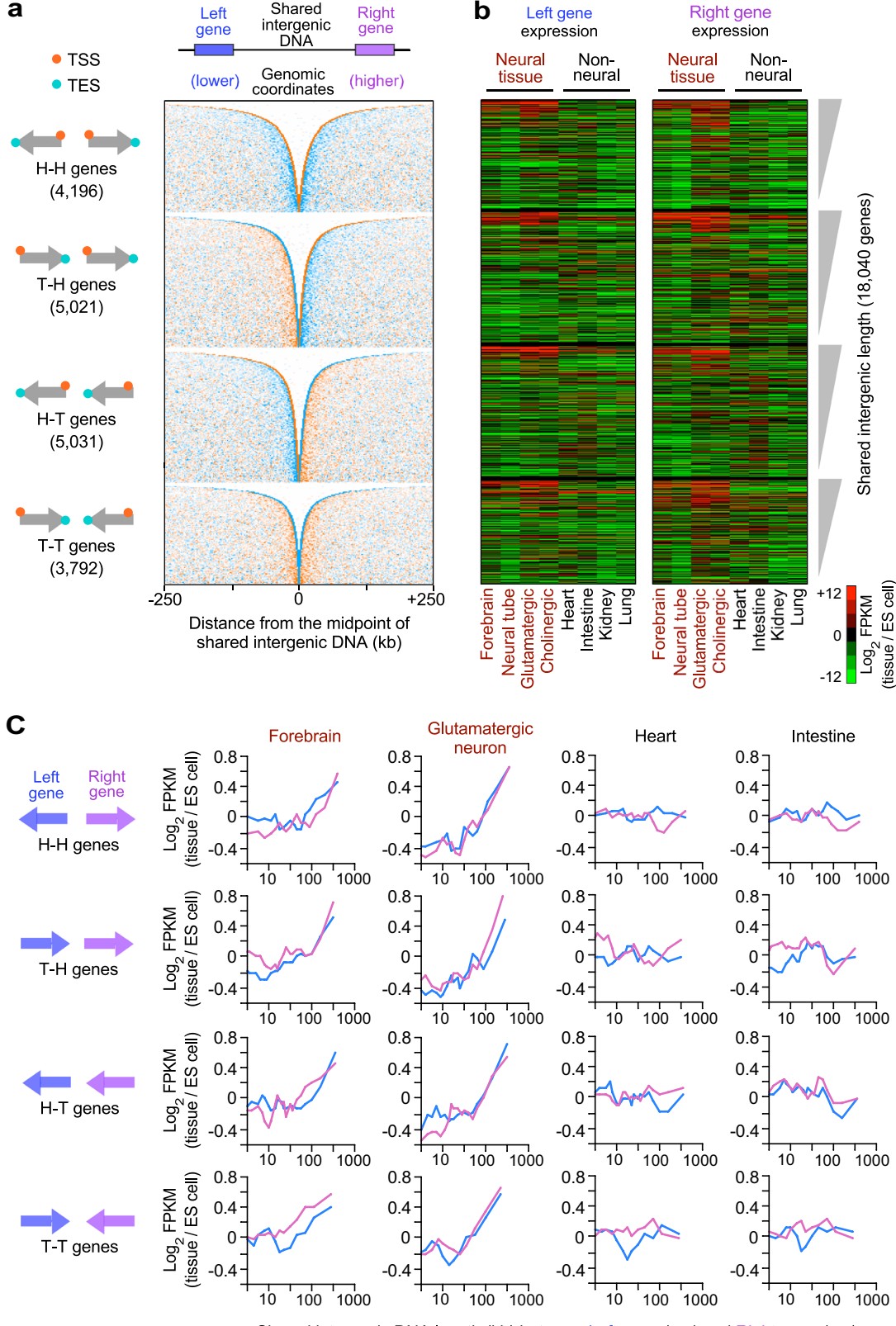

in non-neural tissues in all 4 groups ($P < 0.01$; Supplementary Fig. 9). Moreover, gene ontology analysis revealed that neighboring genes with extended intergenic regions were significantly enriched in neural gene annotations in all 4 groups (Supplementary Fig 10). Our findings demonstrated that the expression of neural genes with extended intergenic regions was globally co-activated with their neighboring neural genes in the nervous system regardless of their gene orientations.

**Neighboring neural genes are controlled by distinct intergenic enhancers.** Our findings suggested that enhancers in long shared

**Fig. 6 Genomic view of tissue-specific expression of neighboring genes depending on intergenic lengths. a** Genome-wide distribution of TSSs (orange dots) and TESs (cyan dots) of 18,040 all mRNA genes, sorted by the shared intergenic lengths between neighboring genes, in the 4 groups of gene pairs. Arrows show the transcription direction of gene pairs. **b** Gene induction levels of the left and right genes in mouse tissues (log$_2$ fold-change in FPKM in each tissue compared to ES cells), ordered by the shared intergenic lengths of the 4 groups shown in (**a**). **c** Genome-wide induction levels of the neighboring left and right genes in the representative tissues, depending on the shared intergenic DNA length. Gene induction levels in each tissue compared to ES cells (log$_2$ fold-change in FPKM) were shown in the 4 groups of gene pairs defined in (**a**). The lines indicate the mean gene induction levels for genes, binned according to the shared intergenic lengths for each tissue (200 genes per bin). All gene pairs and log$_2$ FKPM values used in (**a**, **b**, **c**) are listed in Supplementary Data 13.

intergenic DNA regulate the expression of neighboring neural genes. Consequently, we asked the important question of whether enhancers in extended intergenic regions are necessary for the expression of only one or both of the neuron-specific neighboring genes. To answer this question, we performed loss of function tests using CRISPR/Cas9-based deletions of intergenic enhancers between neighboring neural genes in mouse spinal motor neurons. We selected five motor neuron enhancers, identified in a previous study[23], which are bound by the motor neuron TF Isl1, enriched for both ChIP-seq H3K27ac and ATAC-seq signals (Fig. 7a–c, Supplementary Fig. 11a, b). These five enhancers reside in the shared intergenic regions between two neighboring neural genes, such as *Ntf3* (Neurotrophin)/*Kcna5* (Potassium channel protein) and *Chrm2* (Cholinergic receptor)/*Ptn* (Neurite outgrowth factor), which are expressed specifically in spinal motor neurons. We generated five mouse ES cell lines, each containing the deletion of one intergenic enhancer, which were then differentiated into spinal motor neurons[23]. 4 of the 5 tested enhancer deletions resulted in a significant decrease in the expression of only one neighboring gene compared to wild type motor neurons (Fig. 7d, e, Supplementary Fig. 11c, d). 1 of the 5 tested enhancer deletions showed a significant reduction of expression levels for both neighboring genes compared to wild type motor neurons (Fig. 7f). Since there are multiple enhancers in the extended intergenic regions between two genes, each intergenic enhancer mainly regulates the expression of one neighboring gene but both neighboring genes might be co-activated by multiple enhancers in extended intergenic regions. Indeed, we observed that each of the neighboring neural genes, *Ntf3* and *Kcna5*, required distinct enhancers for their gene expression in motor neurons (NK-E1 enhancer for *Ntf3* expression, and NK-E2 enhancer for *Kcna5* expression, Fig. 7d, Supplementary Fig. 11d). Together, our results indicated that the majority of neighboring neural genes are regulated by independent enhancers rather than by common enhancers in extended shared intergenic regions, although a small portion of intergenic enhancers can co-regulate the expression of both neighboring genes.

Next, we investigated whether enhancers in extended intergenic regions interact with only one or both neighboring genes in neurons and neural tissues. To examine long-range chromatin interactions between intergenic enhancers and promoters of genes, we used Hi-C (genomic chromosome conformation capture assay) datasets of neurons and neural tissues[58,59]. Our Hi-C analyses in cortical neurons identified 1254 intergenic enhancers of EID genes (intergenic lengths >500 kb) that interact with 818 genes (Supplementary Table 1). We found that 72.2% of these enhancers (905 out of 1254) interacted with the TSS of one of the neighboring genes via long-range chromatin interactions in mouse cortical neurons (Fig. 7g). Only 4.5% of these enhancers (57 out of 1254) interacted with the TSSs of both neighboring genes. Our Hi-C analyses in the human cortex and hippocampus also showed that 60–73% of the intergenic enhancers of EID genes interacted with one neighboring gene, whereas <3% of the intergenic enhancers of EID genes interacted with both

neighboring genes. These findings indicated that most enhancers in the long shared intergenic regions interact with only one neighboring gene in neurons and neural tissues. These results are supported with the implication that CTCF binding in the long intergenic regions of neural genes may be associated with TAD boundaries (Supplementary Fig. 8a), which restrict interactions of enhancers to only one neighboring gene promoter.

Genes with high regulatory complexity are induced by a variety of cellular stimuli[60]. In response to differential stimuli in the nervous system, activity-dependent genes or early-response genes are rapidly induced by activation of stimulus–dependent enhancers[61,62]. Recent studies have revealed that the expression of activity-dependent genes in mouse cortical neurons is differentially regulated by the distinct intergenic enhancers upon treatment of different stimuli, such as potassium chloride (KCl)-mediated membrane depolarization and brain-derived neurotrophic factor (BDNF)[63,64]. However, it is still not known whether neighboring genes containing multiple activity-dependent enhancers are co-regulated in response to different stimuli in the nervous system. To address this question, we examined whether activity-dependent genes in cortical neurons were co-activated with their neighboring genes in response to neural stimuli. We cultured primary neurons isolated from the embryonic (E16.5) mouse cortex, then applied different stimulation. We examined whether the expression of the previously reported early-response genes[63,64], such as *c-Fos*, *b-Jun*, and *Arc*, was activated in cortical neurons in response to KCl stimulation. Of the early-response genes, 72.2% (13 out of 18) were induced by more than 2-fold upon KCl stimulation, whereas <6% of their neighboring genes (2 out of 36) were co-activated by more than 2-fold upon KCl stimulation (Supplementary Fig. 12a, Supplementary Data 16). We further compared the expression levels of genes highly induced by differential stimuli in cortical neurons: 202 genes induced by KCl, 118 genes by BDNF, and 170 genes by forskolin, with their neighboring genes. We found that most of the neighboring genes, both upstream and downstream, flanked by activity-dependent genes (focal genes), were not co-activated upon stimulation by KCl, BDNF, and forskolin (Supplementary Fig. 12b). This suggests that stimulus-type-dependent enhancers contribute mainly to the expression of one gene (focal gene), rather than co-activating another neighboring gene upon stimulation. Altogether, our analyses support the conclusion that the majority of neuronal enhancers in extended intergenic regions regulate the expression of only one neighboring neural gene in the mammalian nervous system.

**Intergenic length-dependent neural gene enrichment in vertebrates.** Our finding of the positive relationship between intergenic DNA lengths and gene induction levels in the mammalian nervous system raises the question of whether this phenomenon is also present in other multicellular organisms. Therefore, we performed gene ontology analyses for the top 5% of the genes with the longest intergenic regions of all the protein-coding genes in the chickens, zebrafish, insects, and nematodes. We found that

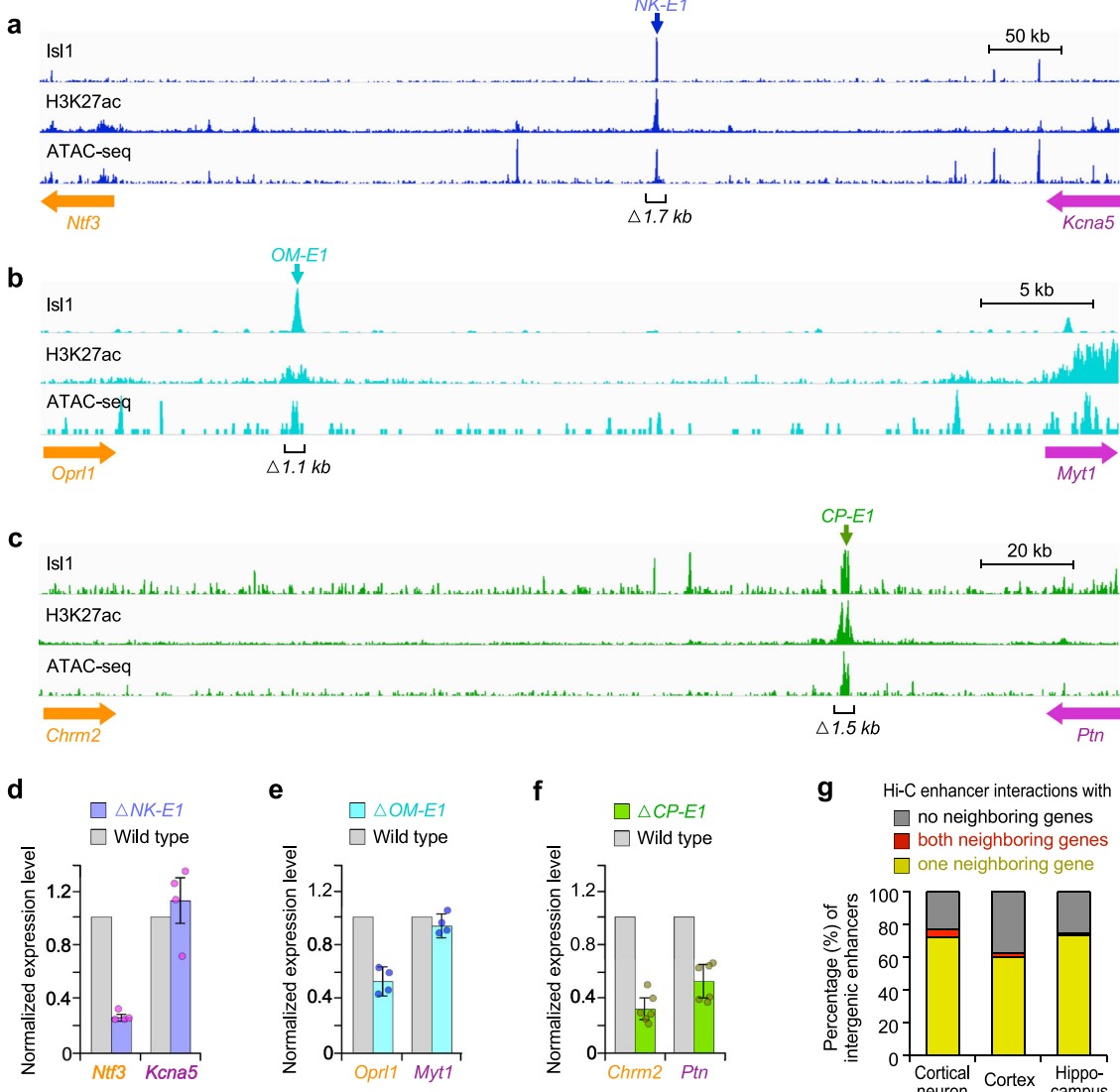

**Fig. 7 Intergenic enhancers are necessary for the expression of neighboring neural genes in motor neurons. a** The location of Isl1-bound intergenic enhancers, detected by ChIP-seq Isl1, ChIP-seq H3K27ac, and ATAC-seq, located between *Ntf3* and *Kcna5* genes in motor neurons differentiated from mouse ES cells. 1719 bp genomic DNA of motor neuron-specific NK-E1 enhancer (blue arrow) was deleted in an ES cell using CRISPR genome editing. **b**, **c** Same as (**a**) except OM-E1 (cyan arrow) and CP-E1 (green arrow) enhancers that are located in the intergenic regions between *Oprl1* and *Myt1*, and *Chrm2* and *Ptn* genes, respectively. 1109 bp of OM-E1 and 1468 bp of CP-E1 enhancers were deleted. **d** Expression levels of neighboring genes, *Ntf3* and *Kcna5*, measured by quantitative RT-PCR in motor neurons containing the NK-E1 enhancer deletion, normalized to wild type expression levels of *Ntf3* and *Kcna5*. Error bars represent standard deviation (SD). Data are presented as mean values $+/-$ SD. Dots represent the corresponding data points (Supplementary Data 14). Two biologically independent differentiated cells were examined over 2 independent experiments ($n = 4$). **e** Same as (**d**) except expression levels of *Oprl1* and *Myt1* genes for OM-E1 enhancer deletion. Two biologically independent differentiated cells were examined over 2 independent experiments ($n = 4$). **f** Same as (**d**) except expression levels of *Chrm2* and *Ptn* genes for CP-E1 enhancer deletion. Two biologically independent differentiated cells were examined over 3 independent experiments ($n = 6$). **g** Percentage of intergenic enhancers interacting with the promoters of the neighboring genes detected by Hi-C. The number of interactions between enhancers and promoters in the mouse cortical neurons, human cortex, and hippocampus was shown in Supplementary Table 1 and Data 15.

the genes with long intergenic regions in *Gallus gallus* and *Danio rerio* were significantly enriched in neural gene annotations, such as nervous system development and neurogenesis (Fig. 8a). In contrast, the genes with long intergenic regions in *D. melanogaster* and *C. elegans* were not enriched in neural gene annotations (Fig. 8b). We also observed that the genes with long intergenic regions in mice were enriched with neural gene annotations (Fig. 3c). Our results indicated that the association of neural genes with long intergenic regions is unique to vertebrates.

We next investigated the genome-wide relationship between intergenic DNA lengths and gene induction levels in various tissues and cell types in *Drosophila* using available RNA-seq datasets (Supplementary Data 1). We found that the top 5% highly induced genes in the *Drosophila* brain and neurons (495 genes), out of all the mRNA genes, did not have long intergenic regions compared to those in non-neural tissues and cells (Fig. 8c, Supplementary Fig. 13a). Rather, the intergenic lengths of highly induced genes in the *Drosophila* brain and neurons were similar to those in non-neural tissues and cells (Fig. 8d). We also found that highly induced genes in terminally differentiated cells and tissues, such as neurons, legs, wings, and testis, on average have longer intergenic regions than undifferentiated or less specified

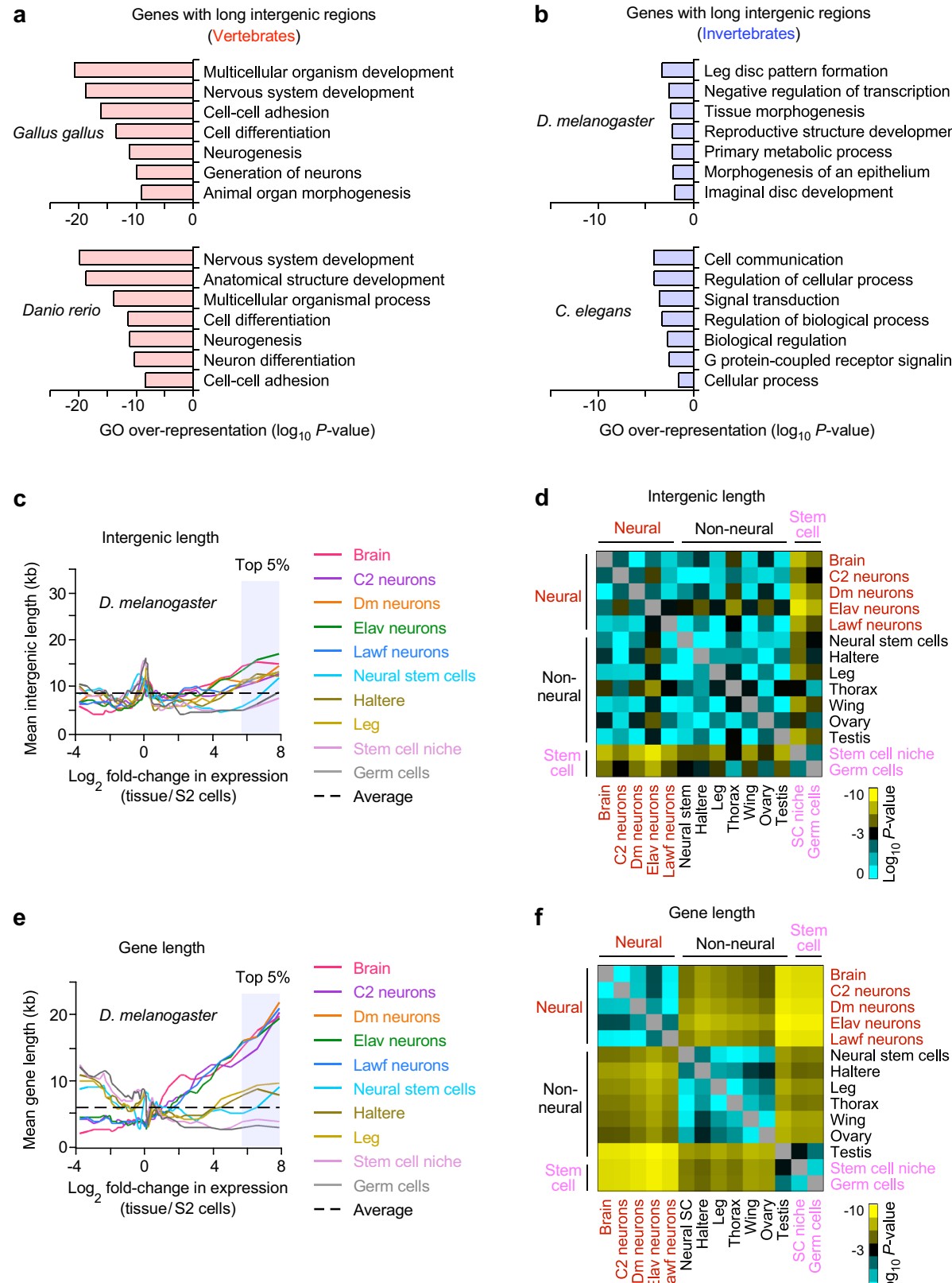

cells (stem cell niche and germ cells). The genes with long intergenic regions were enriched with gene annotations for tissue patterning and morphogenesis (Fig. 8b). Our observations are corroborated by the previous finding that developmentally regulated genes with regulatory complexity are flanked by longer intergenic regions than functionally simple genes in *Drosophila*[8].

Noncoding DNA in *Drosophila* is significantly enriched within genic regions compared to intergenic regions, whereas noncoding DNA in mammals is enriched more in intergenic regions than genic regions[65–67]. We hypothesized that gene induction levels in the *Drosophila* nervous system might be associated more with the length of the gene, containing a large portion of noncoding DNA,

**Fig. 8 Long intergenic regions are enriched in neural genes only in vertebrates. a, b** Gene ontology analysis for the top 5% genes, having the longest intergenic regions, of all the protein-coding genes in each organism. The most enriched seven non-redundant GO terms were shown (Supplementary Data 6, 7). A one-sided binomial test was used to calculate $P$-values in gene set enrichment analysis. $P$-values were not adjusted for multiple comparisons. **c** Genome-wide relationship between intergenic DNA lengths and gene induction levels, assessed by RNA-seq analysis of tissues and cells in *Drosophila melanogaster* (Supplementary Data 1). The lines represent the mean intergenic lengths for genes binned according to the gene induction level for each tissue compared to embryonic Schneider 2 (S2) cells ($\log_2$ fold-change in FPKM, 200 genes per bin, 9897 mRNA genes, Supplementary Data 18). The dotted line indicates the average intergenic length of all the mRNA genes (8.3 kb). **d** Heatmap of the $P$-values (two-sided, two-sample $t$-test) for the difference in intergenic lengths of the top 5% highly induced genes in each tissue (495 genes), out of all the mRNA genes, between pairwise combinations of the representative tissues shown in Supplementary Fig. 13. $P$-values were not adjusted for multiple comparisons. Yellow represents significant $P$-values. Cyan and black represent less or non-significant $P$-values ($P > 1 \times 10^{-3}$) (Supplementary Data 3). **e, f** Same analyses as shown in (**c, d**) except gene lengths instead of intergenic lengths. Gene length is defined as the distance between transcription start and end sites. The average gene length (6.6 kb) of all the mRNA genes, shown as the dotted line in (**e**), was used for normalization to the average intergenic length shown in (**c**).

rather than intergenic lengths. We found that highly induced genes in *Drosophila* neural tissues and neurons as a population had significantly longer gene lengths than genes highly induced in non-neural tissues ($P < 10^{-6}$; Fig. 8e, f). Highly induced genes in the *Drosophila* brain and neurons had at least two times longer median gene lengths than highly induced genes in other tissues (brain: 9.0 kb, *Elav* neurons: 14.5 kb, non-neural tissues: 1.3–4.5 kb; Supplementary Fig. 13b). These results suggest that neuron-specific gene expression relies more on regulatory DNA elements in genic regions than those in intergenic regions in *Drosophila*.

In addition, we examined whether long gene lengths were associated with neural genes in other organisms. We found that the long genes in *Gallus gallus* and *Danio rerio* were significantly enriched with neural gene annotations (Supplementary Fig. 14a). In contrast, the long genes in *C. elegans*, which has fewer neural cell types than *Drosophila*[68,69], were not enriched in neural gene annotations (Supplementary Fig. 14b). In addition, we observed that the long genes in mice and humans were enriched with neural gene annotations (Supplementary Fig. 14c), as reported in other studies[70,71]. Our results suggest that the gene length-dependent neural gene enrichment is associated with the complexity of the nervous system of organisms. Taken together, our findings indicate that the neuron-specific expression of genes with long intergenic regions is unique to vertebrates, which have evolved higher-level cognitive functions compared to invertebrates. The size of the noncoding DNA within and around genes may reflect the levels of regulatory information required to produce complex, cell-type-specific gene expression patterns in the vertebrate nervous system.

## Discussion

We showed that genes with extended intergenic DNA were enriched in neural gene annotations, and were preferentially expressed in mammalian neural tissues, whereas non-neural tissues repress the expression of these genes. An extended intergenic region is an important feature of the neuron-specific gene enrichment that we observed throughout the vertebrate nervous system but not in invertebrates including *D. melanogaster* and *C. elegans*. Neural genes are often flanked by other neural genes that share extended intergenic regions, suggesting that regulatory elements in long shared intergenic regions are available to both neighboring neural genes. Most of the intergenic enhancers, tested in this study, regulate one neighboring neural gene in motor neurons. An intergenic enhancer may regulate one or another neighboring gene upon differential stimuli in the nervous system.

We found that genes with extended intergenic regions contain twice the number of accessible chromatin regions and active enhancers in their long intergenic DNA in neural tissues compared with other tissues, thus allowing for the neuron-specific activation of genes. The diverse and complex gene expression

patterns in the mammalian nervous system must have high temporal and spatial tissue specificity[61,72]. Neural genes may exploit high levels of regulatory information in their extended intergenic regions to control the regulatory complexity of their gene expression in the mammalian nervous system. We also found that highly induced genes, in both neurons and non-neural astrocytes and oligodendrocytes, have extended intergenic regions. This finding further suggests that these glial cell-specific genes may require a large number of regulatory elements to support the complex regulatory functions of neurons in mice and humans.

As found in the previous studies[12,13], we observed that genes highly induced in hematopoietic and immune cells had a high number of accessible DNA regions in their intergenic regions, reflecting the regulatory complexity of gene expression in these terminally differentiated cells. However, long intergenic length-dependent gene induction patterns are unique to neural tissues and neurons, compared with non-neural tissues and cells that we examined in mammals. We found that intergenic enhancers of highly induced genes in neural cells and tissues were significantly dispersed throughout the long intergenic regions, whereas those in non-neural cells and tissues were likely to be densely spaced within short intergenic regions. What would be the potential advantages of having long intergenic regions around neural genes? Long intergenic regions may provide more genomic spaces for diverse neural gene expression patterns in the complex mammalian nervous system.

We propose that generic neural genes expressed in multiple neural tissues have extended intergenic regions. These extended intergenic regions contain the distinct enhancers involved in the expression of neighboring neural genes in a tissue-specific manner. As a result, spatially and temporally diverse patterns of gene expression can be achieved in the complex nervous system (Fig. 9). To increase the regulatory complexity of gene expression, generic neural genes require extended noncoding intergenic regions, containing an average of 8 potential tissue-specific enhancers per intergenic region (Supplementary Fig. 8d). These enhancers are dispersed in long intergenic regions (13.6–16.8 kb between the closest intergenic enhancers per neural gene; Supplementary Fig. 5). Long intergenic regions of neural genes also have a large number of cell- and tissue-specific enhancers bound by the distinct neural TFs, reflecting high levels of regulatory information to accommodate the complex gene expression patterns found in the nervous system. Tissue-specific neural TFs may be involved in enhancer activation and chromatin structure changes by recruiting coactivators and architectural proteins. We observed that an architectural protein, CTCF, was globally bound to commonly accessible DNA sites of neural genes in all the tissues and cell types examined here. CTCF may be involved in separating intergenic *cis*-regulatory elements into two compartments for the independent regulation of two neighboring genes.

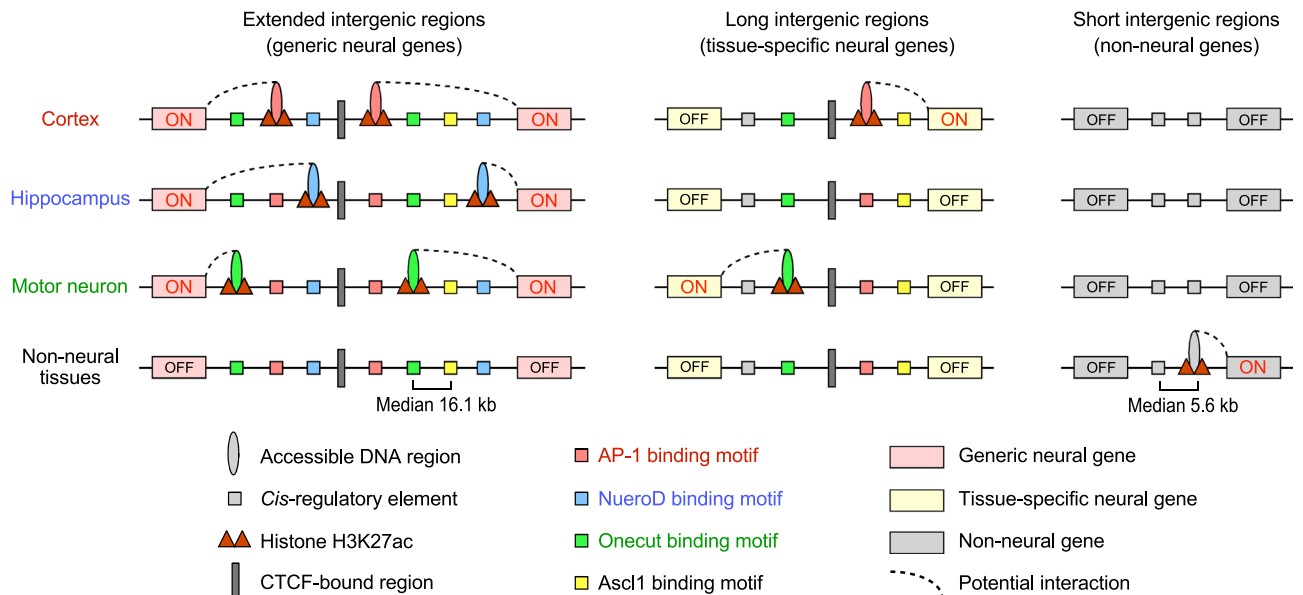

**Fig. 9 Extended intergenic regions contain distinct tissue-specific enhancers contributing to the expression of neighboring neural genes.** We propose that generic neural genes expressed in multiple neural tissues have extended intergenic regions (median length: 180.7 kb) with an average of 8 neural tissue-specific enhancers (median 16.1 kb between the closest enhancers per intergenic region; Supplementary Figs. 5, 8d). Tissue-specific neural genes also have significantly longer intergenic regions than non-neural genes. Neural genes mostly share their intergenic regions with neighboring neural genes, having many cell- and tissue-specific intergenic enhancers bound by the distinct neural TFs. These neural TFs sites are chromatin accessible and enriched with acetylation of H3K27 in a tissue-specific manner. An insulator binding protein, CTCF, binds to commonly accessible DNA sites in the intergenic regions. As a result, co-expression of neighboring neural genes is controlled mostly by distinct intergenic enhancers rather than common enhancers. We propose that a large number of neural cis-regulatory elements in extended intergenic regions lead to tissue- and developmental-specific neural gene expression in the complex mammalian nervous system.

How the long intergenic regions of neural genes are involved in long-range chromatin interactions between enhancers and neighboring genes remains to be examined.

Controlling the complexity and diversity of neural expression patterns requires a large number of regulatory elements in non-coding DNA[73]. We found that neural genes in mammals are flanked by substantially extended intergenic regions, which contain many accessible DNA regions and potential enhancers. Epigenetic regulatory information, such as methylation of DNA, also exists within noncoding DNA regions. Recent studies have found that long genes, containing many methylated DNA regions within their genic regions, are preferentially expressed in the mouse brain relative to other tissues[70,71]. Long genes have more potential enhancers within a genic DNA region than short genes. Long genes might also be associated with diverse gene expression patterns in the vertebrate nervous system. It remains to be determined whether and how neural genes utilize the comprehensive regulatory information, including epigenetic marks and chromatin structure, within and around neural genes to produce neuronal complexity. Long intergenic regions may provide more genomic spaces for higher-order chromatin structures in the mammalian nervous system. Dispersed regulatory elements in long intergenic regions may allow chromatin regulators or architectural proteins, that orchestrate three-dimensional genome organization and TADs, to facilitate more complex gene regulation via long-range chromatin interactions. Taken together, our findings suggest that noncoding DNA located around neural genes is organized to form a unique genomic architecture, in which the expansion of regulatory DNA accommodates complex gene expression patterns in the mammalian nervous system.

## Methods

**Ethical statements**. This study complies with all relevant ethical regulations at the University of Toronto. Mouse studies were conducted strictly following all relevant ethical regulations and standard operating procedures in the animal use protocol (Protocol No. 20012209), which was approved by the Biological Sciences Local Animal Care Committee (LACC) at the University of Toronto, complying with guidelines established by the University of Toronto Animal Care Committee and the Canadian Council on Animal Care.

**Cell culture**. The iCre mouse embryonic stem cell (ESC) line was obtained from Hynek Wichterle (Columbia University)[74]. For motor neuron differentiation, iCre ES cells were plated in ADFNK medium (10% Knockout Serum Replacement (Gibco, 10828028), 2 mM L-glutamine, 0.1 mM 2-mercaptoethanol in 1:1 ratio of Advanced DMEM/F12 (Gibco, 12634028) and Neurobasal medium (Gibco, 21103049)) to initiate the formation of embryoid bodies (day 0). Patterning of embryoid bodies was induced by supplementing media on day 2 with 1 μM all-trans retinoic acid (Sigma, R2625) and 0.25 μM Smoothened agonist (SMO) of hedgehog signaling (Selleckchem, S7779). 5 μM DAPT (Selleckchem, S2215) was added to the ADFNK medium on day 4 and day 5. Differentiated motor neurons were collected on day 6[23].

**Generation of ESC lines containing enhancer deletion**. The iCre ESC line was used for the CRISPR/Cas9 genome editing. Guide RNAs (gRNAs) were designed with the available software[75]. All gRNA sequences used in this study are listed in Supplementary Methods. To generate ESC lines containing enhancer deletion, ESCs were seeded and incubated for 48 h on mouse embryonic fibroblast cells (Millipore, EmbryoMax, Cat No. PMEF-N) to maintain stem cell pluripotency. Cells were transiently transfected with a pair of pgRNA vectors and pCas9-GFP plasmid (Addgene, 44719) using Mouse P3 Primary Cell Nucleofector Kit (Lonza). ESCs were dissociated 24–48 h after transfection, GFP-expressing ESCs were selected by fluorescence-activated cell sorting (FACS). In 4–6 days after FACS sorting and plating, a single ESC colony was picked, followed by genotyping using PCR primers around genomic regions of intended enhancer deletion[23].

**Mouse embryos and cortical neuron stimulation**. The C57BL/6 mouse strains (*Mus musculus*) were purchased from the Charles River Laboratories (Wilmington, MA, USA). Male C57BL/6 mice (aged 10 to 12 weeks) were mated with female mice (aged 8 to 10 weeks). Mice were bred and maintained at 19–23 °C with 40–60% humidity on a fixed 12-h light and 12-h dark cycle in which lights on at 7 am and lights off at 7 pm. E16.5 embryonic mouse cortices were dissected from pregnant female mice, dissociated at 37 °C for 20 min in papain dissociation solution, and stopped by adding 20% of horse serum (Gibco, 26050070) and 80% of Neurobasal medium (Gibco, 21103049). Dissociated neurons were plated and

cultured on cell culture dishes in Neurobasal and B27 medium (Gibco, 12587001) containing 0.5 mM L-glutamine (Gibco, 25030081) and 1x penicillin/streptomycin (Gibco, 15140122) at 37 °C in a $CO_2$ incubator (5%). For potassium chloride (KCl) stimulation, cortical neurons cultured for 5 days in vitro (DIV) were treated with 1 μM tetrodotoxin (TTX; Abcam, ab120055) and 100 μM DL-AP5 (Sigma, A8054) overnight, and then changed into Neurobasal/B27 medium containing 31% of KCl depolarization buffer (170 mM KCl, 2 mM $CaCl_2$, 1 mM $MgCl_2$, 10 mM HEPES) for 2 h stimulation. For BDNF and forskolin stimulation, DIV6 cortical neurons were treated with 10 ng/ml BDNF (R&D System, 248-BDB) or 10 μM forskolin (Sigma, 344270) for 1 h.

**RNA-seq.** Total RNA was extracted from day 6 of differentiated motor neurons using TRIzol reagent (Invitrogen, 15596026). NEBNext Poly(A) mRNA Magnetic Isolation Kit (NEB, E7490) was used to remove ribosomal RNA followed by DNase treatment. Total RNA for primary cortical neurons was extracted after 6 days in vitro culture using TRIzol reagent. RNA-seq libraries were prepared for sequencing using NEBNext RNA Library Prep Kit (NEB, E7765) and were sequenced by Illumina HiSeq 2500 and NovaSeq 6000 System.

**ChIP-seq.** Approximately 25 million cells were crosslinked with 1% formaldehyde and then lysed. Chromatin pellets were isolated, solubilized, and then fragmented by sonication. Fragmented chromatin was then subjected to immunoprecipitation using magnetic beads coupled with Dynabeads Protein A (Invitrogen, 10001D) and antibodies (~3 μg) against the protein of interest (H3K27ac; Abcam, 4729, CTCF; Millipore, 07-729) in 1 mL of 0.5% bovine serum albumin in phosphate-buffered saline solution. After washing beads to remove unbound proteins and DNA, ChIP-seq samples were eluted from the magnetic beads, reverse crosslinked, and then purified. ChIP-seq libraries were prepared for sequencing using NEBNext DNA Library Prep Kit (NEB, E7645) and were sequenced by Illumina HiSeq 2500 and Illumina Nova Seq 6000 System.

**Datasets.** Published mouse RNA-seq, ATAC-seq, and ChIP-seq datasets were downloaded from the ENCODE project website (www.encodeproject.org) and the previous studies[18,23,66,76]. Human RNA-seq datasets were obtained from the ENCODE project website and the previous studies[67,77]. Published RNA-seq, ATAC-seq, ChIP-seq, and Hi-C datasets were downloaded from the ENCODE project website (www.encodeproject.org) and the previous studies, reported in Supplementary Data 1.

**Processing of sequencing data.** For next-generation sequencing data analysis, adapter sequences were removed using Trim Galore (version 0.4), downloaded from Babraham Bioinformatics Institute (http://www.bioinformatics.babraham.ac.uk/projects/trim_galore/). For RNA-seq analysis, the raw trimmed reads were aligned to the mouse reference genome using the STAR aligner (version 2.7), downloaded from GitHub (https://github.com/alexdobin/STAR)[78]. The gene annotation file (release M25, GRCm38.p6, comprehensive gene annotation file) was downloaded from GENCODE (www.gencodegenes.org). Then, fragments per kilobase of transcript per million reads (FPKM) were generated with the Cufflinks package (version 2.2.1), which was conducted using the Galaxy tools (https://usegalaxy.org/)[79]. For ATAC-seq or ChIP-seq analysis, the raw trimmed reads were aligned to the mouse genome with Bowtie2 aligner (version 2.4.2) using the Galaxy tools. Binding locations enriched by ChIP-seq H3K27ac were identified with the MACS2 peak caller (version 2.1.1) using Galaxy ChIP-seq tools.

**Defining intergenic lengths.** 19,144 of the annotated known mRNA genes (mm9 NCBI Build 37 RefSeq Genes) were used to define intergenic DNA lengths for most intergenic length analyses in mice. To define intergenic DNA lengths in human datasets, 19,775 of the annotated protein-coding genes were used (hg19 Ensembl Genes). The intergenic length of a gene was calculated as the sum of the upstream and downstream DNA distance from the end of a gene to the end of the nearest protein-coding genes. We calculated the intergenic lengths for the following genes having non-overlapped intergenic regions and FPKM values: 18,383 mRNA genes in mice (mm9) and 19,268 mRNA genes in humans (hg19). To calculate intergenic lengths in other organisms, all the protein-coding genes in *Gallus gallus* (GRCg6a/galGal6), *Danio rerio* (GRCz11/danRer11), *D. melanogaster* (BDGP Release 6/dm6), and *C. elegans* (WBcel235/ce11) were used. Intergenic lengths of all the mRNA genes in mice, humans, *Drosophila*, and other organisms are listed in Supplementary Data 2, 4, 6, and 18.

**Gene expression analysis.** FPKM values were used as the gene expression of an mRNA transcript, measured by RNA-seq. When comparing gene expression between multiple tissues, FPKM values were median normalized across all tissues using a median FPKM value of all the protein-coding genes per tissue as a normalizer. Prior to converting FPKM to logarithmic scales, all FPKM values were added by 0.1 to avoid producing no value in logarithmic scales with zero (log 0). Gene induction levels were calculated by the $\log_2$ fold-change in expression (FPKM) between the target gene and the reference gene. To plot gene induction or expression level against intergenic lengths, we sorted genes from the longest to the

shortest intergenic lengths, then calculate the mean $\log_2$ fold-change in expression or the mean $\log_{10}$ expression within each bin (bin size: 200 genes, 91 bins, 18,383 genes). A moving average of 3 bins is used to smooth the irregularities of each bin.

**SNP analysis.** To examine single nucleotide polymorphisms (SNPs) associated with Parkinson's disease (PD) genetic risk, we used the published SNP datasets for PD risk in humans[30]. This study reported a total of 23,918 SNPs associated with PD genetic risk (PD SNPs). 14,854 out of these 23,918 SNP nucleotides were located within 1 kb to each other. After excluding the SNPs overlapped within 1 kb, we identified 9064 SNP locations (1 kb width). We further filtered out 4604 SNPs that were located within genic regions of 19,268 mRNA genes, having non-overlapped intergenic regions in humans (hg19). We identified 4460 SNP locations to examine their intergenic lengths.

**Gene ontology (GO) analysis.** GO term enrichment analyses were performed using the Gene Ontology Enrichment Analysis and Visualization system (version 2013Mar, http://cbl-gorilla.cs.technion.ac.il/)[80] and the PANTHER classification system (version 14.0, http://geneontology.org/)[81]. To remove redundant GO terms, we used the Reduce and Visualize Gene Ontology software (version 2021Nov, http://revigo.irb.hr/)[82]. Default parameters were used for the GO over-representation test: GO terms that were enriched above the background with a binomial test. In Fig. 3, all the annotated protein-coding genes except olfactory receptor genes were used as the background GO dataset (Supplementary Methods). In Supplementary Figs. 3 and 8, all annotated genes expressed in at least 1 tissue or cell type were used as the background GO dataset. In Fig. 8 and Supplementary Fig. 14, all the annotated unique genes in each organism were used as the background GO dataset. All GO target genes and background genes are listed in Supplementary Data 6.

**Identifying ChIP-seq and ATAC-seq peaks.** Genomic regions enriched by accessible DNA (ATAC-seq peaks) and ChIP-seq peaks were identified using the peak calling algorithm, Model-based Analysis of ChIP-Seq (MACS, version 2.1.1)[83]. The midpoint of the start and end position of the ATAC-seq peaks and H3K27ac-enriched peaks was used as a peak coordinate. For the published ChIP-seq and ATAC-seq datasets from the ENCODE datasets, we used the processed peak calls as H3K27ac-enriched and DNA-accessible regions. In Fig. 4, Active enhancers were defined as H3K27ac-enriched peaks whose genomic regions are located more than 3 kb from the nearest transcription start site (TSS) of known protein-coding genes. In Fig. 5, if the ATAC-seq peaks in each neural tissue resided within 1 kb to each other in all 4 neural tissues, they were defined as commonly accessible DNA sites. ChIP-seq peak calls of H3K27ac and CTCF in motor neurons (BED file format) were reported in Supplementary Data 12.

**Motif enrichment analysis.** We used the Multiple Em for Motif Elicitation (MEME, version 5.4.1) algorithm for the de novo motif discovery (https://meme-suite.org/meme/tools/meme)[46]. We searched the 6–12 bp of enriched motifs occurring proximal to accessible DNA regions within 50 bp from the midpoints of the identified ATAC-seq peaks. The known DNA motifs matched with the de novo DNA motifs were searched against a transcription factor binding DNA motif database (JASPAR CORE, version 2022, https://jaspar.genereg.net/)[84] using the motif comparison tool Tomtom (version 5.4.1, https://meme-suite.org/meme/tools/tomtom)[85]. The target motifs were compared using the Pearson correlation coefficient with $E$-value <1.0e−10. The top-matched motifs with the most significant $P$-value were reported. For the target motif occurrence analysis, we used the Find Individual Motif Occurrences (FIMO, version 5.4.1, https://meme-suite.org/meme/tools/fimo)[86]. 100 bp from the midpoints of the identified ATAC-seq peaks in Fig. 5 were searched for the occurrence of the target motifs on both DNA strands with the matched $P$-value <1.0e−05 (Supplementary Methods). All Minimal MEME Motif Formats for the target DNA motifs are available in Supplementary Data 11.

**Hi-C analysis.** To examine long-range chromatin interactions between intergenic enhancers and TSS of the gene in neural cells and tissues, we used the published Hi-C dataset for mouse cortical neurons[58]. We used mm10 mouse assembly and gene annotation for the Hi-C analysis (mouse mm10 NCBI RefSeq Genes, Dec. 2011 assembly). The Bonev et al. study reported a total of 18,094 interactions between enhancers and TSSs in cortical neurons. Among them, we selected 16,429 interactions involved with protein-coding genes, then identified 8238 interactions that are associated with intergenic enhancers. 2214 out of 8238 interactions were involved with genes with extremely long intergenic DNA (>500 kb, EID genes). Most enhancers interact with multiple TSSs, and we found 1254 enhancers were involved in 2214 interactions with TSSs of 818 genes (Supplementary Data 15). Long-range chromatin interactions between intergenic enhancers and EID gene promoters in the human cortex and hippocampus[59] were analyzed in the same way and reported in Supplementary Table 1.

**Significance test.** The comparison of intergenic DNA lengths or the number of peaks of highly induced genes between cell or tissue types is reported as $P$-value

heatmap plots. The P-values for the difference between tissues were calculated based on a Student's t-distribution under the null hypothesis (two-tailed, two-sample t-test). We used the null hypothesis because we expect no significant difference between two populations or no difference between certain characteristics of a population. All P-values for the differences in intergenic lengths (Fig. 1c, f), the number of ATAC-seq peaks (Fig. 4c), and ChIP-seq peaks (Fig. 4f) between pairwise combinations of cell or tissue types used in the heatmap plots are listed in Supplementary Data 3 and 9.

**Statistics and reproducibility.** Sample sizes for RNA-seq, ATAC-seq, and ChIP-seq have been determined empirically to provide statistically sufficient datasets for analysis in this study. Statistical methods were not used to predetermine sample size. No data were excluded from the analyses. All experiments with replicates were repeated independently and reproduced successfully. Randomization was used for the experiments involving mouse embryos. Randomization was not applied to in vitro cell line experiments, because the covariates in motor neuron differentiation are not relevant due to the same genetic background and known genotypes of the cell lines used under the same cell culture condition. Blinding was not applied to the experiments in this study, because all experiments are controlled by genotypes and processed simultaneously. For quantitative data, the average values and variances from biological replicates or independent experiments were shown. The error bars in the box plots represent standard deviation and P-values were calculated by Student's t-tests, as indicated in the figure legends.

**Reporting summary.** Further information on research design is available in the Nature Research Reporting Summary linked to this article.

## Data availability

The raw RNA-seq datasets for cortical neuron experiments (Supplementary Fig. 12) have been deposited in the National Center for Biotechnology Information (NCBI) Sequence Read Archive (SRA) under the study number "SRP272183". The processed RNA-seq datasets for cortical neuron experiments are included in Supplementary Data 17. The RNA-seq and ChIP-seq datasets (raw and processed) for motor neuron experiments (Figs. 1, 3, 4, 5, and 7) have been deposited in the NCBI Gene Expression Omnibus (GEO) under the accession number "GSE154532" and "GSE196170". All other relevant data supporting the key findings of this study are available within the article and its Supplementary Information files or from the corresponding author upon reasonable request.

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

## Acknowledgements

We thank R. Alexandrova (SickKids Hospital) for computational assistance and the members of the Rhee laboratory. We thank H. Wichterle (Columbia University) for invaluable support and sharing the iCre ES cell line. We are grateful to J.A. Mitchell (University of Toronto), and M. Closser (Columbia University), and B.F. Pugh (Cornell University) for discussions and comments. The deep sequencing was performed by The Centre for Applied Genomics, Next-Generation Sequencing Facility at The Hospital for Sick Children. The cell sorting was performed by the SickKids-UHN Flow Cytometry Facility. This work was supported by funding from the Natural Sciences and Engineering Research Council of Canada grant RGPIN-2018-06404, Ontario Research Fund Award, John Evans Leader Fund 37547, and the Connaught Scholar Award to H.S.R.

## Author contributions

R.J. and H.S.R. conceived the project. S.-Y.Y. conducted mouse cortical neuron culture and neuron stimulation. S.-Y.Y. and K.P. conducted ChIP-seq experiments. R.J., S.-Y.Y., and K.N.M. performed enhancer deletion experiments. R.J., S.-Y.Y., A.I., and Y.L. performed ES cell culture and RNA-seq experiments. R.J., S.-Y.Y., K.N.M., Z.A., and H.S.R. performed the data analysis. R.J. and H.S.R. wrote the manuscript. All authors read and approved the final manuscript.

## Competing interests

The authors declare no competing interests.

## Additional information

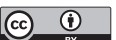

