## [Peer Review File · Nature Communications]

REVIEWER COMMENTS

Reviewer #1 (Remarks to the Author):

In the manuscript, entitled “Extended intergenic DNA contributes to neuron-specific expression of neighboring genes in the mammalian nervous system,” Jaura and colleagues present a series of intriguing observations regarding potential contributions of intergenic DNA length to neural specific gene expression patterns in both human and mouse. By compiling numerous published datasets and comparing to gene expression in ES cells, the authors computationally demonstrate positive associations between intergenic DNA length (i.e., the distance between neighboring genes) and gene expression (displayed as FC vs. ES cell gene expression) in neural (both CNS and PNS) vs. non-neural tissues. They further assess whether intergenic DNA length is associated with the coexpression of neighboring genes, and they identify positive associations in neural, but not in non-neural, tissues. They further posit, based upon overlay assessments with ATAC-seq datasets, that the greater intergenic lengths associated with higher expression of neural vs. non-neural genes likely result from an increased number of accessible active enhancers (H3K27ac marked) within these regions. Presumably these active enhancers are repressed in non-neural tissues. While such observations are interesting, especially in light of a growing literature suggesting neural biases in gene length vs. neural specific gene expression (with these genes being more prone to dysregulation in neurological disorders, such as autism), the manuscript as presented is purely descriptive and lacks necessary mechanistic assessments to truly be able to determine whether such associations are indeed due to the increased number of active enhancers present. Without such assessments, I feel that these data are better suited for a more specialized journal.

1) Given that the authors make a strong point regarding associations between intergenic DNA length and coexpression of neighboring genes in neural tissues – again, presumably regulated by common enhancers –, it would be nice to compare these data to Hi-C datasets to demonstrate that these neighboring genes are truly interacting with common intergenic loci to regulate their expression.

2) Since many genes, such as Fos, in the nervous system contain multiple cognate enhancers that can be differentially regulated depending on stimulus type (e.g., glutamate vs. BDNF stimulations), it would be interesting to know whether neighboring genes containing multiple stimulus-type dependent enhancers are indeed co-regulated based upon differential stimuli.

3) The authors should consider performing targeted manipulations (e.g., CRISPRa/CRISPRi) of intergenic enhancers (either in isolation or multiplexed), followed by assessments of coexpression regulation between neighboring genes. This could be done in cultured neurons to validate their assumptions based upon the presented computational assessments.

4) Given the high number of intergenic SNPs identified in neuronal disorders (e.g., autism, schizophrenia, etc.), could the authors compare their datasets to those from diseased individuals to further illustrate potential roles between intergenic DNA length (and the enhancers contained within these intergenic regions) and neural specific gene expression?

5) Are there common transcription factor binding motifs found in long intergenic regions in neural tissues that may explain commonalities between the types of genes being regulated in this manner?

Reviewer #2 (Remarks to the Author):

The manuscript entitled “Extended intergenic DNA contributes to neuron-specific expression of neighboring genes in the mammalian nervous system” revisits the question of regulatory complexity of cell type specific genes. The authors focus on the regulatory complexity of neuronal specific genes and show that their regulatory complexity is higher than more ubiquitous genes or genes that are non-brain tissue specific. This is a very timely topic given the increasing availability of technologies that are suitable to study regulatory element function.

The authors have found a compelling way to display how gene intergenic space correlates with different features (Figures 1a, 2b, 3b,c, 4a,d) which makes their findings easy to appreciate.

Our main concern with this work is the lack of context.

On the other hand there has been significant work regarding regulatory complexity some of which the authors either ignore. This seriously decreases the potential impact of the manuscript because it leaves many obvious questions unanswered.

1) References 6-10 have explored the relationship between gene intergenic space and tissue specificity. Their conclusion mainly is that there is a correlation between complexity of expression (# of tissues in which a gene is expressed) and intergenic spacing.

2) Recent work has reached similar conclusion as the authors regarding hematopoietic cell differentiation (PMID: 26390058). Their definition of regulatory complexity similar: # of enhancers associated with a gene which they find increased for genes that are induced upon terminal hematopoietic differentiation. Another study found that genes that are early induced by immune stimulus also have higher regulatory complexity (PMID: 29454939).

Given these two points, it is not very surprising that regulatory complexity is also higher for neural specific genes, and it is misleading to suggest that neuronal specific genes are particular in this regard. It is expected given prior in the addressed the importance of intergenic DNA size in regulatory tissue-specific gene expression. It would be useful for the authors to reconcile their observations with these reports.

The authors comparison with tissues and embryonic stem cells does bring an interesting point, which organs, or processes tend to rely on high regulatory complexity vs others that do not. If the authors could shed light onto this question the work would be of greater interest.

Reviewer #3 (Remarks to the Author):

The manuscript by Jaura et al. describes the observation that neuronal-specific genes are statistically more likely to be present adjacent to long stretches of intergenic DNA. Using publicly available bulk and single-cell RNA-sequencing data, the authors investigate the composition of these long intergenic DNA regions and postulate that precise neuronal gene regulation requires such intergenic regions.

Overall, I find the main finding to be interesting, unexpected, and likely biologically meaningful. However, I feel that most of the figures in the manuscript are used to restate the same finding rather than expand upon it. It would seem that this paper could easily be condensed into two figures, one describing the initial observation and one showing the enrichment of neuronal-specific putative regulatory elements in the intergenic regions. I do have concerns about the presentation of data, the clarity of the findings, and the depth of the work as outlined below.

Major Comments:

1) The key findings of the manuscript are based on Figure 1a. Given the growing number of snRNA-seq studies in mouse and human brain, I would like to see this same analysis performed on individual cell types, including glial cells, and across both organisms. The glial cell comparison seems extremely important to me. Is this phenomenon a brain phenomenon or a neuronal phenomenon? Additionally, the wealth of human data from GTEx should enable the authors to quickly assess how widespread this phenomenon is across the diversity of human tissues (far more than the 8 presented in the manuscript).

2) I think one of the key missed opportunities in this paper is to define how widespread this observation is across different organisms. Is this restricted to mammals for example? What if this observation was part of how higher-level cognitive function evolved? Public datasets certainly exist from other model organisms that would help in addressing this point.

3) The discussion of “focal genes” is very unclear. The diagram in Fig. 2a needs to be improved and the explanation of the analysis needs to be made clearer. Moreover, it would seem to me that the correct way to do this analysis is with correlation of the expression of distal gene + focal gene vs proximal gene + focal gene across different lengths of shared intergenic DNA. This correlation is what would seem important and would imply co-regulation by the intergenic regulatory elements. Given Fig. 1a, it seems unsurprising that, on average, both sides of the long intergenic regions show an increase in expression and are enriched for neuronal GO terms. Said in another way, “distal” and “focal” genes are essentially the same class of genes and their definition seems only dependent on orientation.

4) Fig. 3a – I am having a hard time understanding how this analysis is sufficiently different than the analysis in Fig. 1a. It is showing the same phenomenon with the same set of genes, just defined in a slightly different way.

5) The authors make no attempt to use 3D chromosome conformation data to support their primary conclusion that these long intergenic regions are playing a regulatory role. To me this would be an essential analysis to perform and there are various publicly available datasets in mouse and in human that could be used, including cell type-specific maps (for ex. PMID: 31367015 or PMID: 31367015).

6) The language used throughout the manuscript is imprecise. The authors say things like “universal” which has a very specific meaning that is not supported by the results. I’ve outlined some of those instances below but encourage the authors to rigorously check their word choice throughout the manuscript to avoid overstatement of results.

Minor Comments:

7) It would be good to briefly mention in the main text of the manuscript what is considered a “gene” for determination of intergenic length. Depending on the catalog of genes used, this would be different.

8) Supplementary Figure 4 should be in the main figures in my opinion.

9) Related to point 3 above – Line 104-104: “Our findings 104 indicate that a neuronal gene tends to be flanked by another neuronal gene that shares longer intergenic DNA”. I disagree with this interpretation given that, as far as I can tell, the co-expression of focal and distal genes was not analyzed.

10) The motivation in the introduction seems lacking. For example, Line 45-46 “Long intergenic DNA has the potential to contain a larger amount of regulatory information than short intergenic DNA”. This statement seems true solely based on size but the interesting question to me is why aren’t genes evenly spaced throughout the genome and is this inconsistent spacing related to biology? I think a little extra effort in motivating the

11) Please add the publication information (in addition to GEO accession) to Supplementary Tables 1-3.

12) Line 57 – “Can provide an unlimited number of neurons”. The word unlimited is hyperbole.

13) Line 86 – “These results demonstrate that extended intergenic DNA is a universal feature of neuron-specific gene induction throughout the mammalian nervous system.” This is imprecise language. Observing this phenomenon in mice and in humans does not make it universal.

14) Line 125-126 “We also observed that genes with short intergenic DNA were strongly expressed in all tissues and cells including neural tissues”. Another example of imprecise language. You have only shown a handful of tissues and the word “all” is not supported by the results.

15) Line 180 – “Universal” is used again. Please use a more scientifically accurate term.

**** See Nature Research’s author and referees’ website at www.nature.com/authors for information about policies, services and author benefits.**

August 12th, 2021

RE: Responses to Reviewers' Comments on Nature Communications manuscript NCOMMS-20-32976

We thank the three reviewers for their constructive comments and suggestions on the manuscript, entitled “**Extended intergenic DNA contributes to neuron-specific expression of neighboring genes in the mammalian nervous system**”. We took to heart all reviewers’ recommendations and have performed additional experiments and data analyses to address reviewers’ concerns. We have made significant changes to the revised manuscript, which we believe addresses all of the key points raised by the reviewers. We are confident that the revisions made the manuscript more scientifically rigorous and biologically interesting. Below, please find detailed point-by-point responses (in black) to all of the specific comments from the Reviewers (in blue). All of the textual changes in the manuscript are highlighted in yellow.

Reviewers’ comments:

Reviewer #1 (Remarks to the Author):

In the manuscript, entitled “Extended intergenic DNA contributes to neuron-specific expression of neighboring genes in the mammalian nervous system,” Jaura and colleagues present a series of intriguing observations regarding potential contributions of intergenic DNA length to neural specific gene expression patterns in both human and mouse. By compiling numerous published datasets and comparing to gene expression in ES cells, the authors computationally demonstrate positive associations between intergenic DNA length (i.e., the distance between neighboring genes) and gene expression (displayed as FC vs. ES cell gene expression) in neural (both CNS and PNS) vs. non-neural tissues. They further assess whether intergenic DNA length is associated with the coexpression of neighboring genes, and they identify positive associations in neural, but not in non-neural, tissues. They further posit, based upon overlay assessments with ATAC-seq datasets, that the greater intergenic lengths associated with higher expression of neural vs. non-neural genes likely result from an increased number of accessible active enhancers (H3K27ac marked) within these regions. Presumably these active enhancers are repressed in non-neural tissues. While such observations are interesting, especially in light of a growing literature suggesting neural biases in gene length vs. neural specific gene expression (with these genes being more prone to dysregulation in neurological disorders,

such as autism), the manuscript as presented is purely descriptive and lacks necessary mechanistic assessments to truly be able to determine whether such associations are indeed due to the increased number of active enhancers present. Without such assessments, I feel that these data are better suited for a more specialized journal.

1) Given that the authors make a strong point regarding associations between intergenic DNA length and coexpression of neighboring genes in neural tissues – again, presumably regulated by common enhancers, it would be nice to compare these data to Hi-C datasets to demonstrate that these neighboring genes are truly interacting with common intergenic loci to regulate their expression.

Response to Point #1: We thank the reviewer for suggesting this analysis, as it can answer whether genes with long intergenic DNA are coexpressed with their neighboring genes by common intergenic enhancers or distinct enhancers. This is an important question because our findings indicate that long intergenic DNA regions, containing a large number of enhancers, affect the expression of neighboring neuronal genes. To address this question, we performed Hi-C analysis (Point #1), stimulation of mouse cortical neurons followed by RNA-seq (Point #2), and CRISPR-based enhancer deletions (Point #3), as the reviewer suggested.

Regarding Point #1, we examined whether enhancers in long intergenic DNA regions interact with only one or both of neighboring genes in neurons and neural tissues. We used the published Hi-C datasets from mouse cortical neurons and human neural tissues^{1,2}. Our Hi-C analyses in cortical neurons identified 1,254 intergenic enhancers of LID genes (intergenic length > 500 kb) that interact with 818 genes. We found that Only 4.5% of these enhancers (57 out of 1,254) interacted with the TSSs of both neighboring genes. 72.2% of these enhancers (905 out of 1,254) interacted with the transcription start site (TSS) of one of the neighboring genes via long-range chromatin interactions in cortical neurons (**Table 1**, Supplementary Data 9). We showed that most enhancers in the long shared intergenic regions interacted with only one neighboring gene in other neural tissues such as the human cortex and hippocampus. This result was further supported by the below results from stimulation experiments of mouse cortical neurons (Point #2) and enhancer deletion experiments (Point #3).

2) Since many genes, such as Fos, in the nervous system contain multiple cognate enhancers that can be differentially regulated depending on stimulus type (e.g., glutamate vs. BDNF stimulations), it would be interesting to know whether neighboring genes containing multiple stimulus-type dependent enhancers are indeed co-regulated based upon differential stimuli.

Response to Point #2: To address this question, we examined whether activity-dependent genes in cortical neurons were co-activated with their neighboring genes in response to neuronal stimuli. To examine the expression levels of early-response genes upon differential stimuli, we performed RNA-

seq experiments for differentially stimulated cortical neurons and unstimulated neurons. We investigated whether the expression of the previously reported early-response genes^{3,4}, such as *c-Fos*, *b-Jun*, and *Arc*, was activated in cortical neurons in response to KCl stimulation. Of the early-response genes, 72.2% (13 out of 18) were induced by more than two-fold upon KCl stimulation, whereas <6% of their neighboring genes (2 out of 36) were co-activated by more than two-fold upon KCl stimulation (Supplementary Fig. 7a). We compared the expression levels of genes highly induced by differential stimuli in cortical neurons: 202 genes induced by KCl, 118 genes by BDNF, and 170 genes by forskolin, with their neighboring genes. We found that most of the neighboring genes, both upstream and downstream, flanked by activity-dependent genes, were not co-activated upon stimulation by KCl, BDNF, and forskolin (Supplementary Figs. 7b-d). This suggests that stimulus-type-dependent enhancers contribute mainly to the expression of one gene, rather than co-activating another neighboring gene upon stimulation.

3) The authors should consider performing targeted manipulations (e.g., CRISPRa/CRISPRi) of intergenic enhancers (either in isolation or multiplexed), followed by assessments of coexpression regulation between neighboring genes. This could be done in cultured neurons to validate their assumptions based upon the presented computational assessments.

Response to Point #3: We thank the reviewer for suggesting this functional experiment, as it is critical to test whether enhancers in long intergenic DNA regions are necessary for the expression of one or both neuron-specific neighboring genes. To address this question, we performed loss of function tests using CRISPR/Cas9-based deletion of intergenic enhancers between neighboring neuronal genes in mouse spinal motor neurons that were differentiated from embryonic stem cells. We generated five embryonic stem (ES) cell lines containing each homozygous deletion of five distinct intergenic enhancers, followed by spinal motor neuron differentiation of these ES cell lines. These five enhancers reside in the intergenic DNA between two neighboring neuronal genes, which are specifically expressed in spinal motor neurons (**Figs. 6a-c**, Supplementary Figs. 6a, b). We found that four out of five tested enhancer deletions result in a significant decrease in the expression of one neighboring gene in the motor neurons (**Figs. 6d, e**, Supplementary Figs. 6c, d). Only one out of the five tested enhancers showed a significant reduction in the expression levels of both neighbor genes in the motor neurons, compared to the wild type (**Fig. 6f**). Our enhancer deletion experiments showed that most neuronal enhancers in long intergenic DNA regions contribute to the expression of one neighboring gene.

Taken together with the Hi-C analysis (Point #1) and neuron stimulation experiments (Point #2), our findings suggest that a majority of neighboring neuronal genes are largely regulated by independent enhancers rather than by common enhancers in shared intergenic regions. We have

now toned down the associations between intergenic DNA length and co-regulation of neighboring genes in neural tissues.

4) Given the high number of intergenic SNPs identified in neuronal disorders (e.g., autism, schizophrenia, etc.), could the authors compare their datasets to those from diseased individuals to further illustrate potential roles between intergenic DNA length (and the enhancers contained within these intergenic regions) and neural specific gene expression?

Response to Point #4: We agree with the reviewer that it is important to examine the potential association of intergenic DNA length with respect to genetic variations in noncoding regulatory DNA, linked to the genetic risk for neurological disorders. To examine whether genes containing SNPs in their intergenic regions had extended intergenic DNA, we used 9,064 SNPs, identified from genome-wide association studies of Parkinson's disease (PD) patients⁵. Our result showed that a total of 49.2% of SNPs associated with PD risk (4,460 out of 9,064) resided within the intergenic regions of 684 genes. We now show that genes containing these intergenic SNPs associated with PD risk had extended intergenic DNA (median 117 kb), with lengths not significantly different from the intergenic lengths of the highly induced genes in the cortex and excitatory neurons (935 genes, $P > 0.02$; Supplementary Figs. 2c, d). These intergenic SNP-containing genes also had significantly longer intergenic DNA regions than highly induced genes in non-neural tissues ($P < 1 \times 10^{-10}$). We now demonstrated that genes associated with PD contain a large number of SNPs in their extended intergenic DNA. Our results suggest that *cis*-regulatory elements in long intergenic DNA regions are involved in neuronal functions, whose misregulation is associated with neurodegeneration in neurological disorders.

5) Are there common transcription factor binding motifs found in long intergenic regions in neural tissues that may explain commonalities between the types of genes being regulated in this manner?

Response to Point #5: We thank the reviewer for suggesting this analysis to address which transcription factors (TFs) preferentially bind to long intergenic DNA regions in a neural tissue-specific manner. To address this question, we used ATAC-seq datasets to identify neuron-specific accessible DNA regions, which were enriched by ATAC-seq signals in several neural tissues such as the forebrain, hindbrain, and neural tube. We examined DNA sequences at these accessible DNA regions using de novo DNA motif discovery analysis at the genomic level. We identified a repeat of an E-box DNA motif, CAGCTG, at the neuron-specific accessible DNA in long intergenic regions, but not in short intergenic regions (E -value = 4.1×10^{-17} ; Supplementary Fig. 4a). This E-box motif is annotated as the basic helix-loop-helix (bHLH) TF binding motif such as *Ascl1* and *Neurg2* TFs, which play an important role in neurogenesis^{6,7}. Our results indicate that accessible

DNA regions in long intergenic regions are preferentially bound by neuron-specific TFs, whereas ubiquitously expressed TFs tend to bind to intergenic DNA regardless of the length of intergenic regions.

Reviewer #2 (Remarks to the Author):

The manuscript entitled “Extended intergenic DNA contributes to neuron-specific expression of neighboring genes in the mammalian nervous system” revisits the question of regulatory complexity of cell type specific genes. The authors focus on the regulatory complexity of neuronal specific genes and show that their regulatory complexity is higher than more ubiquitous genes or genes that are non-brain tissue specific. This is a very timely topic given the increasing availability of technologies that are suitable to study regulatory element function.

The authors have found a compelling way to display how gene intergenic space correlates with different features (Figures 1a, 2b, 3b,c, 4a,d) which makes their findings easy to appreciate.

Our main concern with this work is the lack of context.

On the other hand there has been significant work regarding regulatory complexity some of which the authors either ignore. This seriously decreases the potential impact of the manuscript because it leaves many obvious questions unanswered.

1) References 6-10 have explored the relationship between gene intergenic space and tissue specificity. Their conclusion mainly is that there is a correlation between complexity of expression (# of tissues in which a gene is expressed) and intergenic spacing.

2) Recent work has reached similar conclusion as the authors regarding hematopoietic cell differentiation (PMID: 26390058). Their definition of regulatory complexity similar: # of enhancers associated with a gene which they find increased for genes that are induced upon terminal hematopoietic differentiation. Another study found that genes that are early induced by immune stimulus also have higher regulatory complexity (PMID: 29454939).

Given these two points, it is not very surprising that regulatory complexity is also higher for neural specific genes, and it is misleading to suggest that neuronal specific genes are particular in this regard. It is expected given prior in the addressed the importance of intergenic DNA size in regulatory tissue-specific gene expression. It would be useful for the authors to reconcile their observations with these reports.

The authors comparison with tissues and embryonic stem cells does bring an interesting point, which organs, or processes tend to rely on high regulatory complexity vs others that do not. If the authors could shed light onto this question the work would be of greater interest.

Response to Reviewer #2: We thank the reviewer for suggesting that we integrate our data with that from recently published studies regarding the regulatory complexity of gene expression and address important questions in this context. We have made significant efforts to address and answer important questions regarding tissue-specific regulatory complexity and emphasized the importance of our results in the context of the previous studies.

We integrated our findings with those of two important studies^{8,9}, regarding the regulatory complexity of hematopoietic cells or immune-response cells. These previous studies showed that regulatory complexity, defined as the number of enhancers of a gene, is higher for genes that are induced upon hematopoietic differentiation or immune stimuli. Multiple studies have been identified densely spaced clusters of enhancers, called super-enhancers, in mammalian cell types^{10,11}. Super-enhancers comprise a large number of TF-binding sites within a relatively short noncoding DNA region (<12.5 kb), and control the expression of cell identity genes^{12,13}. As found in super-enhancers, a short noncoding DNA region may contain a large set of TF-binding sites. Considering this possibility, it is not well understood whether genes with high regulatory complexity require long noncoding DNA regions and why genes are differentially spaced throughout the genome.

As found in the previous studies^{8,9}, we also observed that genes highly induced in hematopoietic and immune cells, as well as neurons, had a high number of accessible DNA regions in their intergenic DNA regions, reflecting the regulatory complexity of gene expression in these terminally differentiated cells. However, highly induced genes in hematopoietic cells and immune cells do not have long intergenic DNA regions, compared to neural cells (Supplementary Fig. 2a). We showed that long intergenic length-dependent gene induction patterns are unique to neural tissues and neurons, compared with non-neural tissues and cells. We also examined whether enhancers of highly induced genes in each tissue were dispersed or clustered in their intergenic DNA regions. We found that enhancers associated with neuronal genes tended to be dispersed throughout the long intergenic DNA, whereas enhancers associated with non-neuronal genes were densely spaced or concentrated within relatively short intergenic DNA regions (Supplementary Fig. 4b). These results explain why the regulatory complexity of gene expression does not always correlate with the lengths of intergenic DNA or noncoding DNA regions.

We now discussed the potential advantages of having long intergenic DNA regions around neuronal genes in the revised manuscript. Long intergenic DNA regions may provide more genomic spaces for higher-order chromatin structure. Dispersed regulatory elements in long intergenic DNA regions may allow chromatin regulators or architectural proteins, that orchestrate three-dimensional genome organization, to facilitate more combinations of long-range chromatin interactions for complex gene regulation. We also addressed these questions in the introduction and included the

potential importance of long intergenic DNA length in neuron-specific gene expression in the discussion.

Reviewer #3 (Remarks to the Author):

The manuscript by Jaura et al. describes the observation that neuronal-specific genes are statistically more likely to be present adjacent to long stretches of intergenic DNA. Using publicly available bulk and single-cell RNA-sequencing data, the authors investigate the composition of these long intergenic DNA regions and postulate that precise neuronal gene regulation requires such intergenic regions. Overall, I find the main finding to be interesting, unexpected, and likely biologically meaningful. However, I feel that most of the figures in the manuscript are used to restate the same finding rather than expand upon it. It would seem that this paper could easily be condensed into two figures, one describing the initial observation and one showing the enrichment of neuronal-specific putative regulatory elements in the intergenic regions. I do have concerns about the presentation of data, the clarity of the findings, and the depth of the work as outlined below.

Major Comments:

1) The key findings of the manuscript are based on Figure 1a. Given the growing number of snRNA-seq studies in mouse and human brain, I would like to see this same analysis performed on individual cell types, including glial cells, and across both organisms. The glial cell comparison seems extremely important to me. Is this phenomenon a brain phenomenon or a neuronal phenomenon? Additionally, the wealth of human data from GTEx should enable the authors to quickly assess how widespread this phenomenon is across the diversity of human tissues (far more than the 8 presented in the manuscript).

Response to Point #1: We agree with the reviewer that it is important to examine whether intergenic length-dependent gene expression patterns are specific for neurons or neural tissues containing neurons and non-neural glial cells. To address this question, we investigated the correlation between intergenic lengths and gene induction levels using single-cell RNA-seq or FACS-sorted glial cells, including astrocytes (Aldh111+, Acsa2+), oligodendrocytes, microglial cells (Cd11b+, Cd45+), and myeloid cells in the mouse brain (**Figs. 2a, b**). We found that highly induced genes in astrocytes and oligodendrocytes, which support axons and synaptic functions, contained significantly long intergenic DNA than non-neural tissues, as was observed in neurons ($P < 1 \times 10^{-11}$; **Figs. 2a, b**). However, there was no relationship between intergenic lengths and gene induction levels in other glial cells such as microglia and myeloid cells, which function as immune and blood

cells in the brain, respectively. Our results showed that intergenic length-dependent gene expression is observed in non-neuronal cells such as astrocytes and oligodendrocytes, which support axons and maintain synaptic functions, in the mammalian central nervous system.

In addition, we now examined the relationship between intergenic lengths and gene induction levels across 26 human tissues and cell types, including glial cells, using published RNA-seq datasets. We confirmed that extended intergenic DNA regions are significantly associated with gene expression in neurons, astrocytes, microglia, and neural tissues across the diverse human cell and tissue types. We have added these results in **Figs. 1d–f** and Supplementary Figs. 2a and b.

2) I think one of the key missed opportunities in this paper is to define how widespread this observation is across different organisms. Is this restricted to mammals for example? What if this observation was part of how higher-level cognitive function evolved? Public datasets certainly exist from other model organisms that would help in addressing this point.

Response to Point #2: We thank the reviewer for suggesting this analysis as it is important to know whether our findings of intergenic length-dependent neuronal expression are restricted to mammals, or whether this phenomenon is widespread in other organisms. We examined the genome-wide relationship between intergenic length and gene expression in 15 tissues and cells in *Drosophila melanogaster* using RNA-seq datasets. We found that highly induced genes in the fly brain and neurons did not have extended intergenic DNA regions compared to those in non-neural tissues and cells (**Fig. 7a**). Rather, the intergenic lengths of highly induced genes in the fly brain and neurons were similar to those in non-neural tissues and cells (**Figs. 7b, c**). We now showed that neuron-specific expression of genes with extended intergenic DNA is unique to mice and humans, not to *Drosophila*. We also performed the same analysis of gene length and tissue-specific gene expression in *Drosophila*. Interestingly, we found that genes more highly induced in fly neural tissues had significantly longer gene lengths than those in non-neural tissues and cells. These findings suggest that the expression of neuronal genes in *Drosophila* relies on the regulatory information within a gene (**Figs. 7d-f**).

3) The discussion of “focal genes” is very unclear. The diagram in Fig. 2a needs to be improved and the explanation of the analysis needs to be made clearer. Moreover, it would seem to me that the correct way to do this analysis is with correlation of the expression of distal gene + focal gene vs proximal gene + focal gene across different lengths of shared intergenic DNA. This correlation is what would seem important and would imply co-regulation by the intergenic regulatory elements. Given Fig. 1a, it seems unsurprising that, on average, both sides of the long intergenic regions show an increase in expression and are enriched for neuronal GO terms. Said in another way, “distal” and

“focal” genes are essentially the same class of genes and their definition seems only dependent on orientation.

Response to Point #3: We repeated the neighboring gene analysis in **Fig. 5a** (original Fig. 2a) and made this analysis and explanation clearer. We removed the middle genes (focal genes) from this analysis and compared two groups of neighboring genes: left and right genes, which are located at the lower and higher genomic coordinates relative to the shared intergenic DNA, respectively. Most left genes with more than 500 kb of shared intergenic DNA regions mostly did not overlap with right genes sharing the same intergenic DNA (Supplementary Data 7). Our gene expression analysis in multiple tissues showed that the expression of genes with long shared intergenic DNA was globally co-activated with the neighboring genes in a neural tissue-specific manner (**Figs. 5a-c**).

4) Fig. 3a – I am having a hard time understanding how this analysis is sufficiently different than the analysis in Fig. 1a. It is showing the same phenomenon with the same set of genes, just defined in a slightly different way.

Response to Point #4: We agree that some parts of **Fig. 3** (original Fig. 3b) are redundant, although we plotted gene induction levels depending on intergenic lengths in original Fig. 3b. We removed the original **Fig. 3b** in the main figures, and moved it to Supplementary 3a. We kept **Fig. 3a** because the illustration in **Fig. 3a** is necessary to explain the definition of LID genes used in the following analyses in the revised **Fig. 3c**.

5) The authors make no attempt to use 3D chromosome conformation data to support their primary conclusion that these long intergenic regions are playing a regulatory role. To me this would be an essential analysis to perform and there are various publicly available datasets in mouse and in human that could be used, including cell type-specific maps (for ex. PMID: 31367015 or PMID: 31367015).

Response to Point #5: We thank the reviewer for suggesting this analysis, as it can test our findings in which enhancers in extended intergenic DNA affected the expression of neighboring neuronal genes. Using Hi-C datasets in mouse cortical neurons and human neural tissues (Supplementary Table 1)^{1,2}, we examined whether the promoters of neighboring genes sharing long intergenic DNA regions interact with distinct enhancers or common enhancers in the shared intergenic DNA in neural tissues (**Table 1**). Our results showed that most enhancers in long shared intergenic DNA regions interacted with only one neighboring gene. This observation further supports our conclusion that the majority of intergenic enhancers regulate the expression of one of two neighboring genes (**Fig. 6**). Please also see **Reviewer #1's Point #1**.

6) The language used throughout the manuscript is imprecise. The authors say things like “universal” which has a very specific meaning that is not supported by the results. I’ve outlined some of those instances below but encourage the authors to rigorously check their word choice throughout the manuscript to avoid overstatement of results.

Response to Point #6: We have now made our language clearer and more precise in the revised manuscript. We removed “universal” and replaced it with more appropriate words according to our results.

Minor Comments:

7) It would be good to briefly mention in the main text of the manuscript what is considered a “gene” for determination of intergenic length. Depending on the catalog of genes used, this would be different.

Response to Point #7: We have added a statement to define a gene, used to determine the intergenic length of a gene, in the main text. We used all protein-coding genes for the determination of intergenic lengths. We excluded genes that are located at the end of chromosomes or within other genes. More details of the definition of intergenic lengths are presented in the **Methods** and Supplementary Methods. We also included information about which gene annotation files to identify mRNA genes in mice (19,144 genes), human (19,775 genes), and *drosophila* (13,943 genes) in the **Methods**. All mRNA genes used in this study are listed in Supplementary Data 1, 3, and 11.

8) Supplementary Figure 4 should be in the main figures in my opinion.

Response to Point #8: We have now moved the previous Supplementary Fig. 4 to the revised main **Fig. 1d**. We also performed additional analyses with humane datasets, including glial cells, and included them in **Fig. 1e** and **Fig. 1f**.

9) Related to point 3 above – Line 104-104: “Our findings 104 indicate that a neuronal gene tends to be flanked by another neuronal gene that shares longer intergenic DNA”. I disagree with this interpretation given that, as far as I can tell, the co-expression of focal and distal genes was not analyzed.

Response to Point #9: We agree that we did not analyze the co-expression of neighboring genes in our initial analysis. We have removed this statement, and toned down the associations between intergenic DNA length and co-expression of neighboring genes in neural tissues. Instead, we have stated that “Our findings suggest that the expression of genes with long shared intergenic DNA is globally co-activated with their neighboring genes in a neural tissue-specific manner.” (Line 271-273)

10) The motivation in the introduction seems lacking. For example, Line 45-46 “Long intergenic DNA has the potential to contain a larger amount of regulatory information than short intergenic DNA”. This statement seems true solely based on size but the interesting question to me is why aren't genes evenly spaced throughout the genome and is this inconsistent spacing related to biology? I think a little extra effort in motivating the

Response to Point #10: We have made the effort to state interesting questions in the introduction in the context of previous studies. We have incorporated the published studies regarding the regulatory complexity of genes^{8,9} into the introduction. We also discussed the regulatory complexity of gene expression and intergenic length/gene density. Since we found that highly induced genes only in neural tissues had significantly longer intergenic DNA regions than non-neural tissues, we tested whether enhancers of highly induced genes in each tissue were dispersed or clustered in their intergenic DNA regions. We found that enhancers associated with neuronal genes are more likely to be dispersed throughout long intergenic DNA regions, whereas enhancers associated with non-neuronal genes are often concentrated within relatively short intergenic DNA regions. Densely spaced clusters of enhancers, called super-enhancers, have been identified in mammalian cell types^{10,11}. Super-enhancers comprise a large number of TF-binding sites within a relatively short noncoding DNA region (<12.5 kb), and control the expression of cell identity genes in many non-neuronal cell types^{12,13}. Thus, highly induced genes in non-neural tissues may also contain a large amount of regulatory information in relatively shorter intergenic DNA or genic regions. We now included this possibility in the revised manuscript. Based on previous studies, we also addressed other important questions, that have not been studied, related to the biological meaning of the long intergenic length of neuron-specific genes in the main text. Please also see **Reviewer #2's points**.

11) Please add the publication information (in addition to GEO accession) to Supplementary Tables 1-3.

Response to Point #11: We have included the publication information in the revised Supplementary Table 1 (original Supplementary Table 1–3).

12) Line 57 – “Can provide an unlimited number of neurons”. The word unlimited is hyperbole.

Response to Point #12: We have changed “an unlimited number of neurons” to “a large number of neurons” because we used a highly efficient large-scale differentiation protocol for producing spinal motor neurons from mouse embryonic stem cells¹⁴. We can routinely differentiate more than 300 million spinal motor neurons within a week for genomic mapping experiments in our laboratory.

13) Line 86 – “These results demonstrate that extended intergenic DNA is a universal feature of neuron-specific gene induction throughout the mammalian nervous system.” This is imprecise language. Observing this phenomenon in mice and in humans does not make it universal.

Response to Point #13: We agree that this statement overgeneralized our observation. We have removed this statement and concluded our results more precisely.

14) Line 125-126 “We also observed that genes with short intergenic DNA were strongly expressed in all tissues and cells including neural tissues”. Another example of imprecise language. You have only shown a handful of tissues and the word “all” is not supported by the results.

Response to Point #14: We removed the imprecise statement “all tissues and cells” and changed it to “all neural and non-neural tissues and cells, examined here.”

15) Line 180 – “Universal” is used again. Please use a more scientifically accurate term.

Response to Point #15: We removed the imprecise statement “a universal feature of neuron-specific gene expression” and changed it to “an important feature of neuron-specific gene expression.”

Rebuttal References

- 1 Bonev, B. *et al.* Multiscale 3D Genome Rewiring during Mouse Neural Development. *Cell* **171**, 557-572 e524, doi:10.1016/j.cell.2017.09.043 (2017).
- 2 Jung, I. *et al.* A compendium of promoter-centered long-range chromatin interactions in the human genome. *Nat Genet* **51**, 1442-1449, doi:10.1038/s41588-019-0494-8 (2019).
- 3 Malik, A. N. *et al.* Genome-wide identification and characterization of functional neuronal activity-dependent enhancers. *Nat Neurosci* **17**, 1330-1339, doi:10.1038/nn.3808 (2014).
- 4 Joo, J. Y., Schaukowitch, K., Farbiak, L., Kilaru, G. & Kim, T. K. Stimulus-specific combinatorial functionality of neuronal c-fos enhancers. *Nat Neurosci* **19**, 75-83, doi:10.1038/nn.4170 (2016).
- 5 Pierce, S. E., Tyson, T., Booms, A., Prah, J. & Coetzee, G. A. Parkinson's disease genetic risk in a midbrain neuronal cell line. *Neurobiol Dis* **114**, 53-64, doi:10.1016/j.nbd.2018.02.007 (2018).
- 6 Wapinski, O. L. *et al.* Rapid Chromatin Switch in the Direct Reprogramming of Fibroblasts to Neurons. *Cell Rep* **20**, 3236-3247, doi:10.1016/j.celrep.2017.09.011 (2017).
- 7 Ma, Y. C. *et al.* Regulation of motor neuron specification by phosphorylation of neurogenin 2. *Neuron* **58**, 65-77, doi:10.1016/j.neuron.2008.01.037 (2008).
- 8 Gonzalez, A. J., Setty, M. & Leslie, C. S. Early enhancer establishment and regulatory locus complexity shape transcriptional programs in hematopoietic differentiation. *Nat Genet* **47**, 1249-1259, doi:10.1038/ng.3402 (2015).
- 9 Donnard, E. *et al.* Comparative Analysis of Immune Cells Reveals a Conserved Regulatory Lexicon. *Cell Syst* **6**, 381-394 e387, doi:10.1016/j.cels.2018.01.002 (2018).
- 10 Whyte, W. A. *et al.* Master transcription factors and mediator establish super-enhancers at key cell identity genes. *Cell* **153**, 307-319, doi:10.1016/j.cell.2013.03.035 (2013).
- 11 Hnisz, D. *et al.* Super-enhancers in the control of cell identity and disease. *Cell* **155**, 934-947, doi:10.1016/j.cell.2013.09.053 (2013).
- 12 Pott, S. & Lieb, J. D. What are super-enhancers? *Nat Genet* **47**, 8-12, doi:10.1038/ng.3167 (2015).
- 13 Wang, X., Cairns, M. J. & Yan, J. Super-enhancers in transcriptional regulation and genome organization. *Nucleic Acids Res* **47**, 11481-11496, doi:10.1093/nar/gkz1038 (2019).
- 14 Rhee, H. S. *et al.* Expression of Terminal Effector Genes in Mammalian Neurons Is Maintained by a Dynamic Relay of Transient Enhancers. *Neuron* **92**, 1252-1265, doi:10.1016/j.neuron.2016.11.037 (2016).

REVIEWER COMMENTS

Reviewer #1 (Remarks to the Author):

In this revised version of the manuscript, entitled "Extended intergenic DNA contributes to neuron-specific expression of neighboring genes in the mammalian nervous system," the authors have done a commendable job in attempting to resolve my previous concerns, which centered primarily on a need to directly test the functional significance of intergenic DNA length in neural vs. non-neural cells/tissues in the mediation of cell-/tissue-type gene expression. In other words, in the previous submission, the authors provided a wealth of descriptive data but had not provided mechanistic insights to explain the relevance of these phenomena. In this resubmission, the authors now provide ample new data in an attempt to address all of my criticisms. However, I fear that all of these new data do little to 'close the loop' in terms of telling us why intergenic DNA length might be important for neural specific gene expression patterns. For example, with their new Hi-C analyses, stimulus-dependent gene expression studies and CRISPR-based manipulation experiments, the authors basically demonstrate that some of the characteristics of long intergenic DNA (e.g., the existence of more enhancers) do not contribute significantly to co-expression of neighboring genes and/or their activity dependent regulation. As such, it seems that the novelty of such descriptive findings is reduced, as it remains unclear whether intergenic DNA length is actually causally associated with neural enriched gene expression or if this simply represents an epiphenomenon that may have little functional significance on its own. Additionally, while I appreciate the authors' inclusion of new MOTIF analyses, it feels like a lost opportunity to not take those data forward and test whether E-BOX TFs, such as Ascl1, etc. are really what matters driving for this correlation between intergenic DNA length and cell-type specific expression.

Because of these reasons, while I continue to find potential merit in these findings, without further mechanistic insights, I feel that this work may be better suited to a more specialized journal.

Reviewer #2 (Remarks to the Author):

The authors addressed my main concern.

There are a few minor points that should be addressed:

- 1) The method section describing the gene ontology analysis did not specify the background used. This should be explicitly stated.
- 2) The analysis of gene expression of the genes flanking large intergenic regions seems to leave open one question (Figure 6b). For example, in figure both left and right genes behave similarly. This is not surprising as there is nothing particular about being at an earlier or later genomic coordinate. Have the authors looked at this same analysis but, in a strand specific way? Is the gene up or down stream of the large intergenic region (up/down stream this time on the strand in which the genes are expressed) have a higher expression? What happens when the genes are expressed from opposite strands? The choice of using the genomic coordinate does not seem the most informative to me.

Reviewer #3 (Remarks to the Author):

The authors have addressed all of my initial concerns.

I have only one new comment given the revisions. The new finding that astrocytes and oligodendrocytes seem to share the long intergenic enrichment is interesting and I'm glad that the authors performed this analysis. It would be helpful if the authors could comment on the similarities and differences between the sets of genes that show this phenomenon in neurons, astrocytes, and oligodendrocytes. One possibility is that it is many of the same genes across all three cell types. However, if the neurons, astrocytes, and oligodendrocytes all have different genes in these long intergenic regions, that would also be important to know.

**** See Nature Research's author and referees' website at www.nature.com/authors for information about policies, services and author benefits.**

February 12, 2022

RE: Responses to Reviewers' Comments on Nature Communications manuscript NCOMMS-20-32976B

We thank the reviewers for their constructive comments and suggestions regarding our manuscript, entitled “**Extended intergenic DNA contributes to neuron-specific expression of neighboring genes in the mammalian nervous system.**” We have considered all reviewers’ recommendations carefully and sought to address each comment through our revisions. Therefore, we are confident that the revisions have made the manuscript more scientifically rigorous and biologically interesting. Below, please find detailed point-by-point responses (in black) to each reviewer’s comments (*in blue italic*). All of the textual changes in the manuscript are highlighted in yellow.

REVIEWER COMMENTS:

Reviewer #1 (Remarks to the Author):

In this revised version of the manuscript, entitled "Extended intergenic DNA contributes to neuron-specific expression of neighboring genes in the mammalian nervous system," the authors have done a commendable job in attempting to resolve my previous concerns, which centered primarily on a need to directly test the functional significance of intergenic DNA length in neural vs. non-neural cells/tissues in the mediation of cell-/tissue-type gene expression. In other words, in the previous submission, the authors provided a wealth of descriptive data but had not provided mechanistic insights to explain the relevance of these phenomena. In this resubmission, the authors now provide ample new data in an attempt to address all of my criticisms. However, I fear that all of these new data do little to 'close the loop' in terms of telling us why intergenic DNA length might be important for neural specific gene expression patterns. For example, with their new Hi-C analyses, stimulus-dependent gene expression studies and CRISPR-based manipulation experiments, the authors basically demonstrate that some of the characteristics of long intergenic DNA (e.g., the existence of more enhancers) do not contribute significantly to co-expression of neighboring genes and/or their activity dependent regulation. As such, it seems that the novelty of such descriptive findings is reduced, as it remains unclear whether intergenic DNA length is actually causally associated with neural enriched gene expression or if this simply represents an epiphenomenon that may have little functional significance on its own. Additionally, while I appreciate the authors' inclusion of new MOTIF analyses, it feels like a lost opportunity to not take those data forward and test whether E-

BOX TFs, such as Ascl1, etc. are really what matters driving for this correlation between intergenic DNA length and cell-type specific expression.

Because of these reasons, while I continue to find potential merit in these findings, without further mechanistic insights, I feel that this work may be better suited to a more specialized journal.

Response to Reviewer #1: We thank you for this comment. In this revision, we focused on providing further mechanistic insights to explain why the length of intergenic DNA regions is important for cell- and tissue-type-specific gene expression patterns.

First, we examined which TF-binding motifs were associated with intergenic-length-dependent gene expression in neural tissues. Using ATAC-seq datasets, we identified differentially accessible DNA sites in the intergenic regions of 625 generic neural genes, which were highly expressed in 4 neural tissues (the postnatal P0 forebrain, adult cortex, adult hippocampus, and embryonic motor neurons) but not expressed in non-neural tissues in mice (**Fig. 5a**). We found that the differentially accessible DNA sites in each neural tissue were significantly enriched with distinct neural TF-binding motifs, such as RFX in the forebrain, NeuroD in the hippocampus, Onecut in motor neurons, and the E-box binding protein AP-1 in the cortex (**Fig. 5b**). Surprisingly, the DNA motifs identified in each neural tissue preferentially occurred in the differentially accessible DNA sites in each tissue but not in other tissues (**Fig. 5d**). In addition, we found a DNA motif for another E-box binding protein Ascl1 (CCAGCTG), which has 1 nucleotide difference from the NeuroD motif (CCAICTG) (**Supplementary Fig. 7**). The Ascl1 motif was found in the differentially accessible DNA sites in all 4 neural tissues. These results suggest that a proneural TF, Ascl1, may be involved in neural gene expression in many tissues, while NeuroD TFs may be associated with the hippocampus-specific gene expression. We also found that the ATAC-seq peaks that are specific in each tissue were enriched with dispersed active enhancers, marked by H3K27ac, in a tissue-specific manner (**Supplementary Fig. 8**). Our results indicate that the generic neural genes have many *cis*-regulatory elements for TF binding in their extended intergenic regions. As a result, various neural TFs may be involved in the activation of tissue- and developmental-specific intergenic enhancers with spatial and temporal tissue specificity.

Because the intergenic regions of the generic neural genes had many dispersed enhancers containing binding sites for tissue-specific TFs (**Supplementary Fig. 8**), we further hypothesized that the neural genes expressed in more tissues should have longer intergenic regions containing more tissue-specific TF-binding sites than the neural genes expressed in fewer tissues. Thus, we evaluated this possibility by comparing the intergenic lengths of the neural genes commonly expressed in multiple neural tissues and genes expressed in fewer neural tissues (**Fig. 5e**). Interestingly, we uncovered a positive correlation between the number of neural tissues expressing

the same set of neural genes and their intergenic DNA lengths (**Fig. 5f**). We found that the genes expressed in more neural tissues had significantly longer intergenic regions than the genes expressed in fewer neural tissues. The genes expressed in only 1 neural tissue (tissue-specific genes) had shorter intergenic lengths than genes expressed in multiple neural tissues but had significantly longer intergenic lengths than non-neural genes. We also found that the neural genes commonly expressed in glutamatergic neurons, astrocytes, and oligodendrocytes had significantly longer intergenic lengths than genes expressed in 1 or 2 cell types in the mouse central nervous system (**Fig. 2e**).

We next tested whether constitutive genes required for basic cellular functions, such as housekeeping genes, also had long intergenic regions like the generic neural genes. We found that housekeeping genes had significantly shorter intergenic lengths than other genes such as tissue-specific genes (**Fig. 5f**), implicating that the correlation between the number of tissues expressing the same set of genes and the intergenic DNA lengths of the genes occurred in neural genes, not in housekeeping genes.

Next, we investigated whether the positive relationship between intergenic DNA lengths and gene induction levels in the nervous system was widespread in other organisms. We found that the genes with long intergenic regions in *Gallus gallus* and *Danio rerio* were strongly enriched with neural genes, but those in *D. melanogaster* and *C. elegans* were not (**Fig. 8**). Our results indicate that the association of neural genes with long intergenic regions is unique to vertebrates.

Previously, we found that an increased number of active enhancers were dispersed in the extended intergenic DNA regions of neuronal genes (**Fig. 4**, Supplementary Fig. 5). In addition, we demonstrated that the genes with long intergenic DNA regions were globally co-expressed with their neighboring neural genes controlled by the distinct enhancers in their shared intergenic regions (**Figs. 6, 7**). In this revision, we also showed that CTCF was bound to commonly accessible DNA sites in the long intergenic regions of most generic neural genes (**Figs. 5b, d**, Supplementary Fig. 8), indicating that CTCF binding in long intergenic regions of neural genes might be involved in insulator function between two neighboring neural genes. Together, we propose that the neural genes expressed in multiple neural tissues have extremely long intergenic regions to utilize the distinct enhancers to express the neighboring neural genes in a tissue-specific manner, allowing spatially and temporally diverse patterns of gene expression (**Fig. 9**). In summary, our results in the revised manuscript further support our conclusion that long intergenic DNA contributes to neuron-specific gene expression patterns and may have functional significance in the vertebrate nervous system. The amount of cell- and tissue-specific *cis*-regulatory elements in the long intergenic regions of neural genes may reflect high levels of regulatory information to accommodate the diverse and complex gene expression patterns in the mammalian nervous system.

Reviewer #2 (Remarks to the Author):

The authors addressed my main concern.

There are a few minor points that should be addressed:

1) The method section describing the gene ontology analysis did not specify the background used. This should be explicitly stated.

Response to Point #1: We now clearly described the background set of genes for our gene ontology analyses in Methods, Supplementary Methods, and Supplementary Data.

2) The analysis of gene expression of the genes flanking large intergenic regions seems to leave open one question (Figure 6b). For example, in figure both left and right genes behave similarly. This is not surprising as there is nothing particular about being at an earlier or later genomic coordinate. Have the authors looked at this same analysis but, in a strand specific way? Is the gene up or down stream of the large intergenic region (up/down stream this time on the strand in which the genes are expressed) have a higher expression? What happens when the genes are expressed from opposite strands? The choice of using the genomic coordinate does not seem the most informative to me.

Response to Point #2: We thank you for this comment. In this revision, we have conducted more detailed analyses to test whether neighboring neuronal genes sharing extended intergenic regions are co-expressed at the genome-wide level. We classified all mRNA genes into 4 groups based on the orientation of neighboring gene pairs: head-to-head (H-H), tail-to-head (T-H), head-to-tail (H-T) genes, and tail-to-tail (T-T) genes (**Fig. 6a**). Then, we examined tissue-specific gene induction levels of the upstream and downstream genes in neural and non-neural tissues (**Fig. 6b**). We found that both upstream and downstream genes sharing extended intergenic DNA had significantly high gene induction levels in neural tissues regardless of their gene orientations (**Fig. 6c**). Our results demonstrated that the expression of neural genes with extended intergenic DNA regions was globally co-activated with their neighboring neural genes. Together, our analyses in the revision further support our conclusion that long intergenic DNA regions contribute to the co-expression of neighboring neural genes in the mammalian nervous system.

Reviewer #3 (Remarks to the Author):

The authors have addressed all of my initial concerns.

I have only one new comment given the revisions. The new finding that astrocytes and oligodendrocytes seem to share the long intergenic enrichment is interesting and I'm glad that the authors performed this analysis. It would be helpful if the authors could comment on the similarities and differences between the sets of genes that show this phenomenon in neurons, astrocytes, and oligodendrocytes. One possibility is that it is many of the same genes across all three cell types. However, if the neurons, astrocytes, and oligodendrocytes all have different genes in these long intergenic regions, that would also be important to know.

Response to Reviewer #3: We thank you for this comment. We are glad that we were able to address the comment. We now tested whether glial-cell-specific genes or neural genes have long intergenic DNA regions. We found that 31.4% of the genes highly induced in each cell type (glutamatergic neuron, astrocytes, and oligodendrocytes) were common in all three cell types (**Figs. 2c, d**). The genes highly induced in all three cell types had significantly long intergenic DNA lengths compared to non-neural genes and were enriched with neural gene ontologies (**Fig. 2e**). The glial-cell-specific genes also had significantly longer intergenic DNA lengths than non-neural genes and had glial-cell-specific gene annotations (**Figs. 2e, f**). Together, we demonstrated that both neural genes and glial-cell-specific genes were enriched around long intergenic regions in the mammalian central nervous system.

REVIEWERS' COMMENTS

Reviewer #1 (Remarks to the Author):

The authors have done an outstanding job of responding to my previous concerns, and I now feel that the resulting manuscript is suitable for publication in Nature Communications.

** See Nature Portfolio's author and referees' website at www.nature.com/authors for information about policies, services and author benefits

REVIEWER COMMENTS

Reviewer #1 (Remarks to the Author):

In the manuscript, entitled “Extended intergenic DNA contributes to neuron-specific expression of neighboring genes in the mammalian nervous system,” Jaura and colleagues present a series of intriguing observations regarding potential contributions of intergenic DNA length to neural specific gene expression patterns in both human and mouse. By compiling numerous published datasets and comparing to gene expression in ES cells, the authors computationally demonstrate positive associations between intergenic DNA length (i.e., the distance between neighboring genes) and gene expression (displayed as FC vs. ES cell gene expression) in neural (both CNS and PNS) vs. non-neural tissues. They further assess whether intergenic DNA length is associated with the coexpression of neighboring genes, and they identify positive associations in neural, but not in non-neural, tissues. They further posit, based upon overlay assessments with ATAC-seq datasets, that the greater intergenic lengths

associated with higher expression of neural vs. non-neural genes likely result from an increased number of accessible active enhancers (H3K27ac marked) within these regions. Presumably these active enhancers are repressed in non-neural tissues. While such observations are interesting, especially in light of a growing literature suggesting neural biases in gene length vs. neural specific gene expression (with these genes being more prone to dysregulation in neurological disorders, such as autism), the manuscript as presented is purely descriptive and lacks necessary mechanistic assessments to truly be able to determine whether such associations are indeed due to the increased number of active enhancers present. Without such assessments, I feel that these data are better suited for a more specialized journal.

1) Given that the authors make a strong point regarding associations between intergenic DNA length and coexpression of neighboring genes in neural tissues – again, presumably regulated by common enhancers –, it would be nice to compare these data to Hi-C datasets to demonstrate that these neighboring genes are truly interacting with common intergenic loci to regulate their expression.

2) Since many genes, such as Fos, in the nervous system contain multiple cognate enhancers that can be differentially regulated depending on stimulus type (e.g., glutamate vs. BDNF stimulations), it would be interesting to know whether neighboring genes containing multiple stimulus-type dependent enhancers are indeed co-regulated based upon differential stimuli.

3) The authors should consider performing targeted manipulations (e.g., CRISPRa/CRISPRi) of intergenic enhancers (either in isolation or multiplexed), followed by assessments of coexpression regulation between neighboring genes. This could be done in cultured neurons to validate their assumptions based upon the presented computational assessments.

4) Given the high number of intergenic SNPs identified in neuronal disorders (e.g., autism, schizophrenia, etc.), could the authors compare their datasets to those from diseased individuals to further illustrate potential roles between intergenic DNA length (and the enhancers contained within these intergenic regions) and neural specific gene expression?

5) Are there common transcription factor binding motifs found in long intergenic regions in neural tissues that may explain commonalities between the types of genes being regulated in this manner?

Reviewer #2 (Remarks to the Author):

The manuscript entitled “Extended intergenic DNA contributes to neuron-specific expression of neighboring genes in the mammalian nervous system” revisits the question of regulatory complexity of cell type specific genes. The authors focus on the regulatory complexity of neuronal specific genes and show that their regulatory complexity is higher than more ubiquitous genes or genes that are non-brain tissue specific. This is a very timely topic given the increasing availability of technologies that are suitable to study regulatory element function. The authors have found a compelling way to display how gene intergenic space correlates with different features (Figures 1a, 2b, 3b,c, 4a,d) which makes their findings easy to appreciate.

Our main concern with this work is the lack of context.

On the other hand there has been significant work regarding regulatory complexity some of which the authors

either ignore. This seriously decreases the potential impact of the manuscript because it leaves many obvious questions unanswered.

1) References 6-10 have explored the relationship between gene intergenic space and tissue specificity. Their conclusion mainly is that there is a correlation between complexity of expression (# of tissues in which a gene is expressed) and intergenic spacing.

2) Recent work has reached similar conclusion as the authors regarding hematopoietic cell differentiation (PMID: 26390058). Their definition of regulatory complexity similar: # of enhancers associated with a gene which they find increased for genes that are induced upon terminal hematopoietic differentiation. Another study found that genes that are early induced by immune stimulus also have higher regulatory complexity (PMID: 29454939).

Given these two points, it is not very surprising that regulatory complexity is also higher for neural specific genes, and it is misleading to suggest that neuronal specific genes are particular in this regard. It is expected given prior in the addressed the importance of intergenic DNA size in regulatory tissue-specific gene expression. It would be useful for the authors to reconcile their observations with these reports.

The authors comparison with tissues and embryonic stem cells does bring an interesting point, which organs, or processes tend to rely on high regulatory complexity vs others that do not. If the authors could shed light onto this question the work would be of greater interest.

Reviewer #3 (Remarks to the Author):

The manuscript by Jaura et al. describes the observation that neuronal-specific genes are statistically more likely to be present adjacent to long stretches of intergenic DNA. Using publicly available bulk and single-cell RNA-sequencing data, the authors investigate the composition of these long intergenic DNA regions and postulate that precise neuronal gene regulation requires such intergenic regions. Overall, I find the main finding to be interesting, unexpected, and likely biologically meaningful. However, I feel that most of the figures in the manuscript are used to restate the same finding rather than expand upon it. It would seem that this paper could easily be condensed into two figures, one describing the initial observation and one showing the enrichment of neuronal-specific putative regulatory elements in the intergenic regions. I do have concerns about the presentation of data, the clarity of the findings, and the depth of the work as outlined below.

Major Comments:

1) The key findings of the manuscript are based on Figure 1a. Given the growing number of snRNA-seq studies in mouse and human brain, I would like to see this same analysis performed on individual cell types, including glial cells, and across both organisms. The glial cell comparison seems extremely important to me. Is this phenomenon a brain phenomenon or a neuronal phenomenon? Additionally, the wealth of human data from GTEx should enable the authors to quickly assess how widespread this phenomenon is across the diversity of human tissues (far more than the 8 presented in the manuscript).

2) I think one of the key missed opportunities in this paper is to define how widespread this observation is across different organisms. Is this restricted to mammals for example? What if this observation was part of how higher-level cognitive function evolved? Public datasets certainly exist from other model organisms that would help in addressing this point.

3) The discussion of "focal genes" is very unclear. The diagram in Fig. 2a needs to be improved and the explanation of the analysis needs to be made clearer. Moreover, it would seem to me that the correct way to do this analysis is with correlation of the expression of distal gene + focal gene vs proximal gene + focal gene across different lengths of shared intergenic DNA. This correlation is what would seem important and would imply co-regulation by the intergenic regulatory elements. Given Fig. 1a, it seems unsurprising that, on average, both sides of the long intergenic regions show an increase in expression and are enriched for neuronal GO terms. Said in another way, "distal" and "focal" genes are essentially the same class of genes and their definition seems only dependent on orientation.

4) Fig. 3a – I am having a hard time understanding how this analysis is sufficiently different than the analysis in Fig. 1a. It is showing the same phenomenon with the same set of genes, just defined in a slightly different way.

5) The authors make no attempt to use 3D chromosome conformation data to support their primary conclusion

that these long intergenic regions are playing a regulatory role. To me this would be an essential analysis to perform and there are various publicly available datasets in mouse and in human that could be used, including cell type-specific maps (for ex. PMID: 31367015 or PMID: 31367015).

6) The language used throughout the manuscript is imprecise. The authors say things like “universal” which has a very specific meaning that is not supported by the results. I’ve outlined some of those instances below but encourage the authors to rigorously check their word choice throughout the manuscript to avoid overstatement of results.

Minor Comments:

7) It would be good to briefly mention in the main text of the manuscript what is considered a “gene” for determination of intergenic length. Depending on the catalog of genes used, this would be different.

8) Supplementary Figure 4 should be in the main figures in my opinion.

9) Related to point 3 above – Line 104-104: “Our findings 104 indicate that a neuronal gene tends to be flanked by another neuronal gene that shares longer intergenic DNA”. I disagree with this interpretation given that, as far as I can tell, the co-expression of focal and distal genes was not analyzed.

10) The motivation in the introduction seems lacking. For example, Line 45-46 “Long intergenic DNA has the potential to contain a larger amount of regulatory information than short intergenic DNA”. This statement seems true solely based on size but the interesting question to me is why aren’t genes evenly spaced throughout the genome and is this inconsistent spacing related to biology? I think a little extra effort in motivating the

11) Please add the publication information (in addition to GEO accession) to Supplementary Tables 1-3.

12) Line 57 – “Can provide an unlimited number of neurons”. The word unlimited is hyperbole.

13) Line 86 – “These results demonstrate that extended intergenic DNA is a universal feature of neuron-specific gene induction throughout the mammalian nervous system.” This is imprecise language. Observing this phenomenon in mice and in humans does not make it universal.

14) Line 125-126 “We also observed that genes with short intergenic DNA were strongly expressed in all tissues and cells including neural tissues”. Another example of imprecise language. You have only shown a handful of tissues and the word “all” is not supported by the results.

15) Line 180 – “Universal” is used again. Please use a more scientifically accurate term.

Reviewers' comments:

Reviewer #1 (Remarks to the Author):

In the manuscript, entitled “Extended intergenic DNA contributes to neuron-specific expression of neighboring genes in the mammalian nervous system,” Jaura and colleagues present a series of intriguing observations regarding potential contributions of intergenic DNA length to neural specific gene expression patterns in both human and mouse. By compiling numerous published datasets and comparing to gene expression in ES cells, the authors computationally demonstrate positive associations between intergenic DNA length (i.e., the distance between neighboring genes) and gene expression (displayed as FC vs. ES cell gene expression) in neural (both CNS and PNS) vs. non-neural tissues. They further assess whether intergenic DNA length is associated with the coexpression of neighboring genes, and they identify positive associations in neural, but not in non-neural, tissues. They further posit, based upon overlay assessments with ATAC-seq datasets, that the greater intergenic lengths

associated with higher expression of neural vs. non-neural genes likely result from an increased number of accessible active enhancers (H3K27ac marked) within these regions. Presumably these active enhancers are repressed in non-neural tissues. While such observations are interesting, especially in light of a growing literature suggesting neural biases in gene length vs. neural specific gene expression (with these genes being more prone to dysregulation in neurological disorders, such as autism), the manuscript as presented is purely descriptive and lacks necessary mechanistic assessments to truly be able to determine whether such associations are indeed due to the increased number of active enhancers present. Without such assessments, I feel that these data are better suited for a more specialized journal.

1) Given that the authors make a strong point regarding associations between intergenic DNA length and coexpression of neighboring genes in neural tissues – again, presumably regulated by common enhancers, it would be nice to compare these data to Hi-C datasets to demonstrate that these neighboring genes are truly interacting with common intergenic loci to regulate their expression.

Response to Point #1: We thank the reviewer for suggesting this analysis, as it can answer whether genes with long intergenic DNA are coexpressed with their neighboring genes by common intergenic enhancers or distinct enhancers. This is an important question because our findings indicate that long intergenic DNA regions, containing a large number of enhancers, affect the expression of neighboring neuronal genes. To address this question, we performed Hi-C analysis (Point #1), stimulation of mouse cortical neurons followed by RNA-seq (Point #2), and CRISPR-based enhancer deletions (Point #3), as the reviewer suggested.

Regarding Point #1, we examined whether enhancers in long intergenic DNA regions interact with only one or both of neighboring genes in neurons and neural tissues. We used the published Hi-C datasets from mouse cortical neurons and human neural tissues^{1,2}. Our Hi-C analyses in cortical neurons identified 1,254 intergenic enhancers of LID genes (intergenic length > 500 kb) that interact with 818 genes. We found that Only 4.5% of these enhancers (57 out of 1,254) interacted with the TSSs of both neighboring genes. 72.2% of these enhancers (905 out of 1,254) interacted with the transcription start site (TSS) of one of the neighboring genes via long-range chromatin interactions in cortical neurons (**Table 1**, Supplementary Data 9). We showed that most enhancers in the long shared intergenic regions interacted with only one neighboring gene in other neural tissues such as the human cortex and hippocampus. This result was further supported by the below results from stimulation experiments of mouse cortical neurons (Point #2) and enhancer deletion experiments (Point #3).

2) Since many genes, such as Fos, in the nervous system contain multiple cognate enhancers that can be differentially regulated depending on stimulus type (e.g., glutamate vs. BDNF stimulations), it would be interesting to know whether neighboring genes containing multiple stimulus-type dependent enhancers are indeed co-regulated based upon differential stimuli.

Response to Point #2: To address this question, we examined whether activity-dependent genes in cortical neurons were co-activated with their neighboring genes in response to neuronal stimuli. To examine the expression levels of early-response genes upon differential stimuli, we performed RNA-seq experiments for differentially stimulated cortical neurons and unstimulated neurons. We investigated whether the expression of the previously reported early-response genes^{3,4}, such as *c-Fos*, *b-Jun*, and *Arc*, was activated in cortical neurons in response to KCl stimulation. Of the early-response genes, 72.2% (13 out of 18) were induced by more than two-fold upon KCl stimulation, whereas <6% of their neighboring genes (2 out of 36) were co-activated by more than two-fold upon KCl stimulation (Supplementary Fig. 7a). We compared the expression levels of genes highly induced by differential stimuli in cortical neurons: 202 genes induced by KCl, 118 genes by BDNF, and 170

genes by forskolin, with their neighboring genes. We found that most of the neighboring genes, both upstream and downstream, flanked by activity-dependent genes, were not co-activated upon stimulation by KCl, BDNF, and forskolin (Supplementary Figs. 7b-d). This suggests that stimulus-type-dependent enhancers contribute mainly to the expression of one gene, rather than co-activating another neighboring gene upon stimulation.

3) The authors should consider performing targeted manipulations (e.g., CRISPRa/CRISPRi) of intergenic enhancers (either in isolation or multiplexed), followed by assessments of coexpression regulation between neighboring genes. This could be done in cultured neurons to validate their assumptions based upon the presented computational assessments.

Response to Point #3: We thank the reviewer for suggesting this functional experiment, as it is critical to test whether enhancers in long intergenic DNA regions are necessary for the expression of one or both neuron-specific neighboring genes. To address this question, we performed loss of function tests using CRISPR/Cas9-based deletion of intergenic enhancers between neighboring neuronal genes in mouse spinal motor neurons that were differentiated from embryonic stem cells. We generated five embryonic stem (ES) cell lines containing each homozygous deletion of five distinct intergenic enhancers, followed by spinal motor neuron differentiation of these ES cell lines. These five enhancers reside in the intergenic DNA between two neighboring neuronal genes, which are specifically expressed in spinal motor neurons (Figs. 6a-c, Supplementary Figs. 6a, b). We found that four out of five tested enhancer deletions result in a significant decrease in the expression of one neighboring gene in the motor neurons (Figs. 6d, e, Supplementary Figs. 6c, d). Only one out of the five tested enhancers showed a significant reduction in the expression levels of both neighbor genes in the motor neurons, compared to the wild type (Fig. 6f). Our enhancer deletion experiments showed that most neuronal enhancers in long intergenic DNA regions contribute to the expression of one neighboring gene.

Taken together with the Hi-C analysis (Point #1) and neuron stimulation experiments (Point #2), our findings suggest that a majority of neighboring neuronal genes are largely regulated by independent enhancers rather than by common enhancers in shared intergenic regions. We have now toned down the associations between intergenic DNA length and co-regulation of neighboring genes in neural tissues.

4) Given the high number of intergenic SNPs identified in neuronal disorders (e.g., autism, schizophrenia, etc.), could the authors compare their datasets to those from diseased individuals to further illustrate potential roles between intergenic DNA length (and the enhancers contained within these intergenic regions) and neural specific gene expression?

Response to Point #4: We agree with the reviewer that it is important to examine the potential association of intergenic DNA length with respect to genetic variations in noncoding regulatory DNA, linked to the genetic risk for neurological disorders. To examine whether genes containing SNPs in their intergenic regions had extended intergenic DNA, we used 9,064 SNPs, identified from genome-wide association studies of Parkinson's disease (PD) patients⁵. Our result showed that a total of 49.2% of SNPs associated with PD risk (4,460 out of 9,064) resided within the intergenic regions of 684 genes. We now show that genes containing these intergenic SNPs associated with PD risk had extended intergenic DNA (median 117 kb), with lengths not significantly different from the intergenic lengths of the highly induced genes in the cortex and excitatory neurons (935 genes, $P > 0.02$; Supplementary Figs. 2c, d). These intergenic SNP-containing genes also had significantly longer intergenic DNA regions than highly induced genes in non-neural tissues ($P < 1 \times 10^{-10}$). We now demonstrated that genes associated with PD contain a large number of SNPs in their extended intergenic DNA. Our results suggest that *cis*-regulatory elements in long intergenic DNA regions are involved in neuronal functions, whose misregulation is associated with neurodegeneration in neurological disorders.

5) Are there common transcription factor binding motifs found in long intergenic regions in neural tissues that may explain commonalities between the types of genes being regulated in this manner?

Response to Point #5: We thank the reviewer for suggesting this analysis to address which transcription factors (TFs) preferentially bind to long intergenic DNA regions in a neural tissue-specific manner. To address this question, we used ATAC-seq datasets to identify neuron-specific accessible DNA regions, which were enriched by ATAC-seq signals in several neural tissues such as the forebrain, hindbrain, and neural tube. We examined DNA sequences at these accessible DNA regions using de novo DNA motif discovery analysis at the genomic level. We identified a repeat of an E-box DNA motif, CAGCTG, at the neuron-specific accessible DNA in long

intergenic regions, but not in short intergenic regions (E -value = 4.1×10^{-17} ; Supplementary Fig. 4a). This E-box motif is annotated as the basic helix-loop-helix (bHLH) TF binding motif such as *Ascl1* and *Neurg2* TFs, which play an important role in neurogenesis^{6,7}. Our results indicate that accessible DNA regions in long intergenic regions are preferentially bound by neuron-specific TFs, whereas ubiquitously expressed TFs tend to bind to intergenic DNA regardless of the length of intergenic regions.

Reviewer #2 (Remarks to the Author):

The manuscript entitled “Extended intergenic DNA contributes to neuron-specific expression of neighboring genes in the mammalian nervous system” revisits the question of regulatory complexity of cell type specific genes. The authors focus on the regulatory complexity of neuronal specific genes and show that their regulatory complexity is higher than more ubiquitous genes or genes that are non-brain tissue specific. This is a very timely topic given the increasing availability of technologies that are suitable to study regulatory element function. The authors have found a compelling way to display how gene intergenic space correlates with different features (Figures 1a, 2b, 3b,c, 4a,d) which makes their findings easy to appreciate.

Our main concern with this work is the lack of context.

On the other hand there has been significant work regarding regulatory complexity some of which the authors either ignore. This seriously decreases the potential impact of the manuscript because it leaves many obvious questions unanswered.

1) References 6-10 have explored the relationship between gene intergenic space and tissue specificity. Their conclusion mainly is that there is a correlation between complexity of expression (# of tissues in which a gene is expressed) and intergenic spacing.

2) Recent work has reached similar conclusion as the authors regarding hematopoietic cell differentiation (PMID: 26390058). Their definition of regulatory complexity similar: # of enhancers associated with a gene which they find increased for genes that are induced upon terminal hematopoietic differentiation. Another study found that genes that are early induced by immune stimulus also have higher regulatory complexity (PMID: 29454939). Given these two points, it is not very surprising that regulatory complexity is also higher for neural specific genes, and it is misleading to suggest that neuronal specific genes are particular in this regard. It is expected given prior in the addressed the importance of intergenic DNA size in regulatory tissue-specific gene expression. It would be useful for the authors to reconcile their observations with these reports.

The authors comparison with tissues and embryonic stem cells does bring an interesting point, which organs, or processes tend to rely on high regulatory complexity vs others that do not. If the authors could shed light onto this question the work would be of greater interest.

Response to Reviewer #2: We thank the reviewer for suggesting that we integrate our data with that from recently published studies regarding the regulatory complexity of gene expression and address important questions in this context. We have made significant efforts to address and answer important questions regarding tissue-specific regulatory complexity and emphasized the importance of our results in the context of the previous studies.

We integrated our findings with those of two important studies^{8,9}, regarding the regulatory complexity of hematopoietic cells or immune-response cells. These previous studies showed that regulatory complexity, defined as the number of enhancers of a gene, is higher for genes that are induced upon hematopoietic differentiation or immune stimuli. Multiple studies have been identified densely spaced clusters of enhancers, called super-enhancers, in mammalian cell types^{10,11}. Super-enhancers comprise a large number of TF-binding sites within a relatively short noncoding DNA region (<12.5 kb), and control the expression of cell identity genes^{12,13}. As found in super-enhancers, a short noncoding DNA region may contain a large set of TF-binding sites. Considering this possibility, it is not well understood whether genes with high regulatory complexity require long noncoding DNA regions and why genes are differentially spaced throughout the genome.

As found in the previous studies^{8,9}, we also observed that genes highly induced in hematopoietic and immune cells, as well as neurons, had a high number of accessible DNA regions in their intergenic DNA regions, reflecting the regulatory complexity of gene expression in these terminally differentiated cells. However, highly induced genes in hematopoietic cells and immune cells do not have long intergenic DNA regions, compared to neural cells (Supplementary Fig. 2a). We showed that long intergenic length-dependent gene induction patterns are unique to neural tissues and neurons, compared with non-neural tissues and cells. We also examined whether enhancers of highly induced genes in each tissue were dispersed or clustered in their intergenic DNA regions. We found that enhancers associated with neuronal genes tended to be dispersed throughout the long intergenic DNA, whereas enhancers associated with non-neuronal genes were densely spaced or concentrated

within relatively short intergenic DNA regions (Supplementary Fig. 4b). These results explain why the regulatory complexity of gene expression does not always correlate with the lengths of intergenic DNA or noncoding DNA regions.

We now discussed the potential advantages of having long intergenic DNA regions around neuronal genes in the revised manuscript. Long intergenic DNA regions may provide more genomic spaces for higher-order chromatin structure. Dispersed regulatory elements in long intergenic DNA regions may allow chromatin regulators or architectural proteins, that orchestrate three-dimensional genome organization, to facilitate more combinations of long-range chromatin interactions for complex gene regulation. We also addressed these questions in the introduction and included the potential importance of long intergenic DNA length in neuron-specific gene expression in the discussion.

Reviewer #3 (Remarks to the Author):

The manuscript by Jaura et al. describes the observation that neuronal-specific genes are statistically more likely to be present adjacent to long stretches of intergenic DNA. Using publicly available bulk and single-cell RNA-sequencing data, the authors investigate the composition of these long intergenic DNA regions and postulate that precise neuronal gene regulation requires such intergenic regions. Overall, I find the main finding to be interesting, unexpected, and likely biologically meaningful. However, I feel that most of the figures in the manuscript are used to restate the same finding rather than expand upon it. It would seem that this paper could easily be condensed into two figures, one describing the initial observation and one showing the enrichment of neuronal-specific putative regulatory elements in the intergenic regions. I do have concerns about the presentation of data, the clarity of the findings, and the depth of the work as outlined below.

Major Comments:

1) The key findings of the manuscript are based on Figure 1a. Given the growing number of snRNA-seq studies in mouse and human brain, I would like to see this same analysis performed on individual cell types, including glial cells, and across both organisms. The glial cell comparison seems extremely important to me. Is this phenomenon a brain phenomenon or a neuronal phenomenon? Additionally, the wealth of human data from GTEx should enable the authors to quickly assess how widespread this phenomenon is across the diversity of human tissues (far more than the 8 presented in the manuscript).

Response to Point #1: We agree with the reviewer that it is important to examine whether intergenic length-dependent gene expression patterns are specific for neurons or neural tissues containing neurons and non-neural glial cells. To address this question, we investigated the correlation between intergenic lengths and gene induction levels using single-cell RNA-seq or FACS-sorted glial cells, including astrocytes (Aldh1l1+, Acsa2+), oligodendrocytes, microglial cells (Cd11b+, Cd45+), and myeloid cells in the mouse brain (**Figs. 2a, b**). We found that highly induced genes in astrocytes and oligodendrocytes, which support axons and synaptic functions, contained significantly long intergenic DNA than non-neural tissues, as was observed in neurons ($P < 1 \times 10^{-11}$; **Figs. 2a, b**). However, there was no relationship between intergenic lengths and gene induction levels in other glial cells such as microglia and myeloid cells, which function as immune and blood cells in the brain, respectively. Our results showed that intergenic length-dependent gene expression is observed in non-neuronal cells such as astrocytes and oligodendrocytes, which support axons and maintain synaptic functions, in the mammalian central nervous system.

In addition, we now examined the relationship between intergenic lengths and gene induction levels across 26 human tissues and cell types, including glial cells, using published RNA-seq datasets. We confirmed that extended intergenic DNA regions are significantly associated with gene expression in neurons, astrocytes, microglia, and neural tissues across the diverse human cell and tissue types. We have added these results in **Figs. 1d-f** and Supplementary Figs. 2a and b.

2) I think one of the key missed opportunities in this paper is to define how widespread this observation is across different organisms. Is this restricted to mammals for example? What if this observation was part of how higher-level cognitive function evolved? Public datasets certainly exist from other model organisms that would help in addressing this point.

Response to Point #2: We thank the reviewer for suggesting this analysis as it is important to know whether our findings of intergenic length-dependent neuronal expression are restricted to mammals, or whether this phenomenon is widespread in other organisms. We examined the genome-wide relationship between intergenic length and gene expression in 15 tissues and cells in *Drosophila melanogaster* using RNA-seq datasets. We found that highly induced genes in the fly brain and neurons did not have extended intergenic DNA regions compared to those in non-neural tissues and cells (**Fig. 7a**). Rather, the intergenic lengths of highly induced genes in the fly brain and neurons were similar to those in non-neural tissues and cells (**Figs. 7b, c**). We now showed that neuron-specific expression of genes with extended intergenic DNA is unique to mice and humans, not to *Drosophila*. We also performed the same analysis of gene length and tissue-specific gene expression in *Drosophila*. Interestingly, we found that genes more highly induced in fly neural tissues had significantly longer gene lengths than those in non-neural tissues and cells. These findings suggest that the expression of neuronal genes in *Drosophila* relies on the regulatory information within a gene (**Figs. 7d-f**).

3) The discussion of “focal genes” is very unclear. The diagram in Fig. 2a needs to be improved and the explanation of the analysis needs to be made clearer. Moreover, it would seem to me that the correct way to do this analysis is with correlation of the expression of distal gene + focal gene vs proximal gene + focal gene across different lengths of shared intergenic DNA. This correlation is what would seem important and would imply co-regulation by the intergenic regulatory elements. Given Fig. 1a, it seems unsurprising that, on average, both sides of the long intergenic regions show an increase in expression and are enriched for neuronal GO terms. Said in another way, “distal” and “focal” genes are essentially the same class of genes and their definition seems only dependent on orientation.

Response to Point #3: We repeated the neighboring gene analysis in **Fig. 5a** (original Fig. 2a) and made this analysis and explanation clearer. We removed the middle genes (focal genes) from this analysis and compared two groups of neighboring genes: left and right genes, which are located at the lower and higher genomic coordinates relative to the shared intergenic DNA, respectively. Most left genes with more than 500 kb of shared intergenic DNA regions mostly did not overlap with right genes sharing the same intergenic DNA (Supplementary Data 7). Our gene expression analysis in multiple tissues showed that the expression of genes with long shared intergenic DNA was globally co-activated with the neighboring genes in a neural tissue-specific manner (**Figs. 5a-c**).

4) Fig. 3a – I am having a hard time understanding how this analysis is sufficiently different than the analysis in Fig. 1a. It is showing the same phenomenon with the same set of genes, just defined in a slightly different way.

Response to Point #4: We agree that some parts of **Fig. 3** (original Fig. 3b) are redundant, although we plotted gene induction levels depending on intergenic lengths in original Fig. 3b. We removed the original **Fig. 3b** in the main figures, and moved it to Supplementary 3a. We kept **Fig. 3a** because the illustration in **Fig. 3a** is necessary to explain the definition of LID genes used in the following analyses in the revised **Fig. 3c**.

5) The authors make no attempt to use 3D chromosome conformation data to support their primary conclusion that these long intergenic regions are playing a regulatory role. To me this would be an essential analysis to perform and there are various publicly available datasets in mouse and in human that could be used, including cell type-specific maps (for ex. PMID: 31367015 or PMID: 31367015).

Response to Point #5: We thank the reviewer for suggesting this analysis, as it can test our findings in which enhancers in extended intergenic DNA affected the expression of neighboring neuronal genes. Using Hi-C datasets in mouse cortical neurons and human neural tissues (Supplementary Table 1)^{1,2}, we examined whether the promoters of neighboring genes sharing long intergenic DNA regions interact with distinct enhancers or common enhancers in the shared intergenic DNA in neural tissues (**Table 1**). Our results showed that most enhancers in long shared intergenic DNA regions interacted with only one neighboring gene. This observation further supports our conclusion that the majority of intergenic enhancers regulate the expression of one of two neighboring genes (**Fig. 6**). Please also see **Reviewer #1's Point #1**.

6) The language used throughout the manuscript is imprecise. The authors say things like “universal” which has a very specific meaning that is not supported by the results. I’ve outlined some of those instances below but

encourage the authors to rigorously check their word choice throughout the manuscript to avoid overstatement of results.

Response to Point #6: We have now made our language clearer and more precise in the revised manuscript. We removed “universal” and replaced it with more appropriate words according to our results.

Minor Comments:

7) It would be good to briefly mention in the main text of the manuscript what is considered a “gene” for determination of intergenic length. Depending on the catalog of genes used, this would be different.

Response to Point #7: We have added a statement to define a gene, used to determine the intergenic length of a gene, in the main text. We used all protein-coding genes for the determination of intergenic lengths. We excluded genes that are located at the end of chromosomes or within other genes. More details of the definition of intergenic lengths are presented in the **Methods** and Supplementary Methods. We also included information about which gene annotation files to identify mRNA genes in mice (19,144 genes), human (19,775 genes), and *drosophila* (13,943 genes) in the **Methods**. All mRNA genes used in this study are listed in Supplementary Data 1, 3, and 11.

8) Supplementary Figure 4 should be in the main figures in my opinion.

Response to Point #8: We have now moved the previous Supplementary Fig. 4 to the revised main **Fig. 1d**. We also performed additional analyses with humane datasets, including glial cells, and included them in **Fig. 1e** and **Fig. 1f**.

9) Related to point 3 above – Line 104-104: “Our findings 104 indicate that a neuronal gene tends to be flanked by another neuronal gene that shares longer intergenic DNA”. I disagree with this interpretation given that, as far as I can tell, the co-expression of focal and distal genes was not analyzed.

Response to Point #9: We agree that we did not analyze the co-expression of neighboring genes in our initial analysis. We have removed this statement, and toned down the associations between intergenic DNA length and co-expression of neighboring genes in neural tissues. Instead, we have stated that “Our findings suggest that the expression of genes with long shared intergenic DNA is globally co-activated with their neighboring genes in a neural tissue-specific manner.” (Line 271-273)

10) The motivation in the introduction seems lacking. For example, Line 45-46 “Long intergenic DNA has the potential to contain a larger amount of regulatory information than short intergenic DNA”. This statement seems true solely based on size but the interesting question to me is why aren’t genes evenly spaced throughout the genome and is this inconsistent spacing related to biology? I think a little extra effort in motivating the

Response to Point #10: We have made the effort to state interesting questions in the introduction in the context of previous studies. We have incorporated the published studies regarding the regulatory complexity of genes^{8,9} into the introduction. We also discussed the regulatory complexity of gene expression and intergenic length/gene density. Since we found that highly induced genes only in neural tissues had significantly longer intergenic DNA regions than non-neural tissues, we tested whether enhancers of highly induced genes in each tissue were dispersed or clustered in their intergenic DNA regions. We found that enhancers associated with neuronal genes are more likely to be dispersed throughout long intergenic DNA regions, whereas enhancers associated with non-neuronal genes are often concentrated within relatively short intergenic DNA regions. Densely spaced clusters of enhancers, called super-enhancers, have been identified in mammalian cell types^{10,11}. Super-enhancers comprise a large number of TF-binding sites within a relatively short noncoding DNA region (<12.5 kb), and control the expression of cell identity genes in many non-neural cell types^{12,13}. Thus, highly induced genes in non-neural tissues may also contain a large amount of regulatory information in relatively shorter intergenic DNA or genic regions. We now included this possibility in the revised manuscript. Based on previous studies, we also addressed other important questions, that have not been studied, related to the biological meaning of the long intergenic length of neuron-specific genes in the main text. Please also see **Reviewer #2's points**.

11) Please add the publication information (in addition to GEO accession) to Supplementary Tables 1-3.

Response to Point #11: We have included the publication information in the revised Supplementary Table 1 (original Supplementary Table 1–3).

12) Line 57 – “Can provide an unlimited number of neurons”. The word unlimited is hyperbole.

Response to Point #12: We have changed “an unlimited number of neurons” to “a large number of neurons” because we used a highly efficient large-scale differentiation protocol for producing spinal motor neurons from mouse embryonic stem cells¹⁴. We can routinely differentiate more than 300 million spinal motor neurons within a week for genomic mapping experiments in our laboratory.

13) Line 86 – “These results demonstrate that extended intergenic DNA is a universal feature of neuron-specific gene induction throughout the mammalian nervous system.” This is imprecise language. Observing this phenomenon in mice and in humans does not make it universal.

Response to Point #13: We agree that this statement overgeneralized our observation. We have removed this statement and concluded our results more precisely.

14) Line 125-126 “We also observed that genes with short intergenic DNA were strongly expressed in all tissues and cells including neural tissues”. Another example of imprecise language. You have only shown a handful of tissues and the word “all” is not supported by the results.

Response to Point #14: We removed the imprecise statement “all tissues and cells” and changed it to “all neural and non-neural tissues and cells, examined here.”

15) Line 180 – “Universal” is used again. Please use a more scientifically accurate term.

Response to Point #15: We removed the imprecise statement “a universal feature of neuron-specific gene expression” and changed it to “an important feature of neuron-specific gene expression.”

Rebuttal References

- 1 Bonev, B. et al. Multiscale 3D Genome Rewiring during Mouse Neural Development. *Cell* 171, 557-572 e524, doi:10.1016/j.cell.2017.09.043 (2017).
- 2 Jung, I. et al. A compendium of promoter-centered long-range chromatin interactions in the human genome. *Nat Genet* 51, 1442-1449, doi:10.1038/s41588-019-0494-8 (2019).
- 3 Malik, A. N. et al. Genome-wide identification and characterization of functional neuronal activity-dependent enhancers. *Nat Neurosci* 17, 1330-1339, doi:10.1038/nn.3808 (2014).
- 4 Joo, J. Y., Schaukowitz, K., Farbiak, L., Kilaru, G. & Kim, T. K. Stimulus-specific combinatorial functionality of neuronal c-fos enhancers. *Nat Neurosci* 19, 75-83, doi:10.1038/nn.4170 (2016).
- 5 Pierce, S. E., Tyson, T., Booms, A., Prah, J. & Coetzee, G. A. Parkinson's disease genetic risk in a midbrain neuronal cell line. *Neurobiol Dis* 114, 53-64, doi:10.1016/j.nbd.2018.02.007 (2018).
- 6 Wapinski, O. L. et al. Rapid Chromatin Switch in the Direct Reprogramming of Fibroblasts to Neurons. *Cell Rep* 20, 3236-3247, doi:10.1016/j.celrep.2017.09.011 (2017).

- 7 Ma, Y. C. et al. Regulation of motor neuron specification by phosphorylation of neurogenin 2. *Neuron* 58, 65-77, doi:10.1016/j.neuron.2008.01.037 (2008).
- 8 Gonzalez, A. J., Setty, M. & Leslie, C. S. Early enhancer establishment and regulatory locus complexity shape transcriptional programs in hematopoietic differentiation. *Nat Genet* 47, 1249-1259, doi:10.1038/ng.3402 (2015).
- 9 Donnard, E. et al. Comparative Analysis of Immune Cells Reveals a Conserved Regulatory Lexicon. *Cell Syst* 6, 381-394 e387, doi:10.1016/j.cels.2018.01.002 (2018).
- 10 Whyte, W. A. et al. Master transcription factors and mediator establish super-enhancers at key cell identity genes. *Cell* 153, 307-319, doi:10.1016/j.cell.2013.03.035 (2013).
- 11 Hnisz, D. et al. Super-enhancers in the control of cell identity and disease. *Cell* 155, 934-947, doi:10.1016/j.cell.2013.09.053 (2013).
- 12 Pott, S. & Lieb, J. D. What are super-enhancers? *Nat Genet* 47, 8-12, doi:10.1038/ng.3167 (2015).
- 13 Wang, X., Cairns, M. J. & Yan, J. Super-enhancers in transcriptional regulation and genome organization. *Nucleic Acids Res* 47, 11481-11496, doi:10.1093/nar/gkz1038 (2019).
- 14 Rhee, H. S. et al. Expression of Terminal Effector Genes in Mammalian Neurons Is Maintained by a Dynamic Relay of Transient Enhancers. *Neuron* 92, 1252-1265, doi:10.1016/j.neuron.2016.11.037 (2016).

REVIEWER COMMENTS

Reviewer #1 (Remarks to the Author):

In this revised version of the manuscript, entitled "Extended intergenic DNA contributes to neuron-specific expression of neighboring genes in the mammalian nervous system," the authors have done a commendable job in attempting to resolve my previous concerns, which centered primarily on a need to directly test the functional significance of intergenic DNA length in neural vs. non-neural cells/tissues in the mediation of cell-/tissue-type gene expression. In other words, in the previous submission, the authors provided a wealth of descriptive data but had not provided mechanistic insights to explain the relevance of these phenomena. In this resubmission, the authors now provide ample new data in an attempt to address all of my criticisms. However, I fear that all of these new data do little to 'close the loop' in terms of telling us why intergenic DNA length might be important for neural specific gene expression patterns. For example, with their new Hi-C analyses, stimulus-dependent gene expression studies and CRISPR-based manipulation experiments, the authors basically demonstrate that some of the characteristics of long intergenic DNA (e.g., the existence of more enhancers) do not contribute significantly to co-expression of neighboring genes and/or their activity dependent regulation. As such, it seems that the novelty of such descriptive findings is reduced, as it remains unclear whether intergenic DNA length is actually causally associated with neural enriched gene expression or if this simply represents an epiphenomenon that may have little functional significance on its own. Additionally, while I appreciate the authors' inclusion of new MOTIF analyses, it feels like a lost opportunity to not take those data forward and test whether E-BOX TFs, such as Ascl1, etc. are really what matters driving for this correlation between intergenic DNA length and cell-type specific expression.

Because of these reasons, while I continue to find potential merit in these findings, without further mechanistic insights, I feel that this work may be better suited to a more specialized journal.

Reviewer #2 (Remarks to the Author):

The authors addressed my main concern.

There are a few minor points that should be addressed:

- 1) The method section describing the gene ontology analysis did not specify the background used. This should be explicitly stated.
- 2) The analysis of gene expression of the genes flanking large intergenic regions seems to leave open one question (Figure 6b). For example, in figure both left and right genes behave similarly. This is not surprising as there is nothing particular about being at an earlier or later genomic coordinate. Have the authors looked at this same analysis but, in a strand specific way? Is the gene up or down stream of the large intergenic region (up/down stream this time on the strand in which the genes are expressed) have a higher expression? What happens when the genes are expressed from opposite strands? The choice of using the genomic coordinate does not seem the most informative to me.

Reviewer #3 (Remarks to the Author):

The authors have addressed all of my initial concerns.

I have only one new comment given the revisions. The new finding that astrocytes and oligodendrocytes seem to share the long intergenic enrichment is interesting and I'm glad that the authors performed this analysis. It would be helpful if the authors could comment on the similarities and differences between the sets of genes that show this phenomenon in neurons, astrocytes, and oligodendrocytes. One possibility is that it is many of the same genes across all three cell types. However, if the neurons, astrocytes, and oligodendrocytes all have different genes in these long intergenic regions, that would also be important to know.

Reviewers' comments:

Reviewer #1 (Remarks to the Author):

In this revised version of the manuscript, entitled "Extended intergenic DNA contributes to neuron-specific expression of neighboring genes in the mammalian nervous system," the authors have done a commendable job in attempting to resolve my previous concerns, which centered primarily on a need to directly test the functional significance of intergenic DNA length in neural vs. non-neural cells/tissues in the mediation of cell-/tissue-type gene expression. In other words, in the previous submission, the authors provided a wealth of descriptive data but had not provided mechanistic insights to explain the relevance of these phenomena. In this resubmission, the authors now provide ample new data in an attempt to address all of my criticisms. However, I fear that all of these new data do little to 'close the loop' in terms of telling us why intergenic DNA length might be important for neural specific gene expression patterns. For example, with their new Hi-C analyses, stimulus-dependent gene expression studies and CRISPR-based manipulation experiments, the authors basically demonstrate that some of the characteristics of long intergenic DNA (e.g., the existence of more enhancers) do not contribute significantly to co-expression of neighboring genes and/or their activity dependent regulation. As such, it seems that the novelty of such descriptive findings is reduced, as it remains unclear whether intergenic DNA length is actually causally associated with neural enriched gene expression or if this simply represents an epiphenomenon that may have little functional significance on its own. Additionally, while I appreciate the authors' inclusion of new MOTIF analyses, it feels like a lost opportunity to not take those data forward and test whether E-BOX TFs, such as Ascl1, etc. are really what matters driving for this correlation between intergenic DNA length and cell-type specific expression. Because of these reasons, while I continue to find potential merit in these findings, without further mechanistic insights, I feel that this work may be better suited to a more specialized journal.

Response to Reviewer #1: We thank you for this comment. In this revision, we focused on providing further mechanistic insights to explain why the length of intergenic DNA regions is important for cell- and tissue-type-specific gene expression patterns.

First, we examined which TF-binding motifs were associated with intergenic-length-dependent gene expression in neural tissues. Using ATAC-seq datasets, we identified differentially accessible DNA sites in the intergenic regions of 625 generic neural genes, which were highly expressed in 4 neural tissues (the postnatal P0 forebrain, adult cortex, adult hippocampus, and embryonic motor neurons) but not expressed in non-neural tissues in mice (**Fig. 5a**). We found that the differentially accessible DNA sites in each neural tissue were significantly enriched with distinct neural TF-binding motifs, such as RFX in the forebrain, NeuroD in the hippocampus, Onecut in motor neurons, and the E-box binding protein AP-1 in the cortex (**Fig. 5b**). Surprisingly, the DNA motifs identified in each neural tissue preferentially occurred in the differentially accessible DNA sites in each tissue but not in other tissues (**Fig. 5d**). In addition, we found a DNA motif for another E-box binding protein Ascl1 (CCAGCTG), which has 1 nucleotide difference from the NeuroD motif (CCAICTG) (Supplementary Fig. 7). The Ascl1 motif was found in the differentially accessible DNA sites in all 4 neural tissues. These results suggest that a proneural TF, Ascl1, may be involved in neural gene expression in many tissues, while NeuroD TFs may be associated with the hippocampus-specific gene expression. We also found that the ATAC-seq peaks that are specific in each tissue were enriched with dispersed active enhancers, marked by H3K27ac, in a tissue-specific manner (Supplementary Fig. 8). Our results indicate that the generic neural genes have many *cis*-regulatory elements for TF binding in their extended intergenic regions. As a result, various neural TFs may be involved in the activation of tissue- and developmental-specific intergenic enhancers with spatial and temporal tissue specificity.

Because the intergenic regions of the generic neural genes had many dispersed enhancers containing binding sites for tissue-specific TFs (Supplementary Fig. 8), we further hypothesized that the neural genes expressed in more tissues should have longer intergenic regions containing more tissue-specific TF-binding sites than the neural genes expressed in fewer tissues. Thus, we evaluated this possibility by comparing the intergenic lengths of the neural genes commonly expressed in multiple neural tissues and genes expressed in fewer neural tissues (**Fig. 5e**). Interestingly, we uncovered a positive correlation between the number of neural tissues expressing the same set of neural genes and their intergenic DNA lengths (**Fig. 5f**). We found that the genes expressed in more neural tissues had significantly longer intergenic regions than the genes expressed in fewer neural tissues. The genes expressed in only 1 neural tissue (tissue-specific genes) had shorter intergenic lengths than genes expressed in multiple neural tissues but had significantly longer intergenic lengths than non-neural genes. We also found that the neural genes commonly expressed in glutamatergic neurons, astrocytes, and oligodendrocytes had significantly longer intergenic lengths than genes expressed in 1 or 2 cell types in the mouse central nervous system (**Fig. 2e**).

We next tested whether constitutive genes required for basic cellular functions, such as housekeeping genes, also had long intergenic regions like the generic neural genes. We found that housekeeping genes had

significantly shorter intergenic lengths than other genes such as tissue-specific genes (Fig. 5f), implicating that the correlation between the number of tissues expressing the same set of genes and the intergenic DNA lengths of the genes occurred in neural genes, not in housekeeping genes.

Next, we investigated whether the positive relationship between intergenic DNA lengths and gene induction levels in the nervous system was widespread in other organisms. We found that the genes with long intergenic regions in *Gallus gallus* and *Danio rerio* were strongly enriched with neural genes, but those in *D. melanogaster* and *C. elegans* were not (Fig. 8). Our results indicate that the association of neural genes with long intergenic regions is unique to vertebrates.

Previously, we found that an increased number of active enhancers were dispersed in the extended intergenic DNA regions of neuronal genes (Fig. 4, Supplementary Fig. 5). In addition, we demonstrated that the genes with long intergenic DNA regions were globally co-expressed with their neighboring neural genes controlled by the distinct enhancers in their shared intergenic regions (Figs. 6, 7). In this revision, we also showed that CTCF was bound to commonly accessible DNA sites in the long intergenic regions of most generic neural genes (Figs. 5b, d, Supplementary Fig. 8), indicating that CTCF binding in long intergenic regions of neural genes might be involved in insulator function between two neighboring neural genes. Together, we propose that the neural genes expressed in multiple neural tissues have extremely long intergenic regions to utilize the distinct enhancers to express the neighboring neural genes in a tissue-specific manner, allowing spatially and temporally diverse patterns of gene expression (Fig. 9). In summary, our results in the revised manuscript further support our conclusion that long intergenic DNA contributes to neuron-specific gene expression patterns and may have functional significance in the vertebrate nervous system. The amount of cell- and tissue-specific *cis*-regulatory elements in the long intergenic regions of neural genes may reflect high levels of regulatory information to accommodate the diverse and complex gene expression patterns in the mammalian nervous system.

Reviewer #2 (Remarks to the Author):

The authors addressed my main concern.

There are a few minor points that should be addressed:

1) The method section describing the gene ontology analysis did not specify the background used. This should be explicitly stated.

Response to Point #1: We now clearly described the background set of genes for our gene ontology analyses in Methods, Supplementary Methods, and Supplementary Data.

2) The analysis of gene expression of the genes flanking large intergenic regions seems to leave open one question (Figure 6b). For example, in figure both left and right genes behave similarly. This is not surprising as there is nothing particular about being at an earlier or later genomic coordinate. Have the authors looked at this same analysis but, in a strand specific way? Is the gene up or down stream of the large intergenic region (up/down stream this time on the strand in which the genes are expressed) have a higher expression? What happens when the genes are expressed from opposite strands? The choice of using the genomic coordinate does not seem the most informative to me.

Response to Point #2: We thank you for this comment. In this revision, we have conducted more detailed analyses to test whether neighboring neuronal genes sharing extended intergenic regions are co-expressed at the genome-wide level. We classified all mRNA genes into 4 groups based on the orientation of neighboring gene pairs: head-to-head (H-H), tail-to-head (T-H), head-to-tail (H-T) genes, and tail-to-tail (T-T) genes (Fig. 6a). Then, we examined tissue-specific gene induction levels of the upstream and downstream genes in neural and non-neural tissues (Fig. 6b). We found that both upstream and downstream genes sharing extended intergenic DNA had significantly high gene induction levels in neural tissues regardless of their gene orientations (Fig. 6c). Our results demonstrated that the expression of neural genes with extended intergenic DNA regions was globally co-activated with their neighboring neural genes. Together, our analyses in the revision further support our conclusion that long intergenic DNA regions contribute to the co-expression of neighboring neural genes in the mammalian nervous system.

Reviewer #3 (Remarks to the Author):

The authors have addressed all of my initial concerns.

I have only one new comment given the revisions. The new finding that astrocytes and oligodendrocytes seem to share the long intergenic enrichment is interesting and I'm glad that the authors performed this analysis. It would be helpful if the authors could comment on the similarities and differences between the sets of genes that show this phenomenon in neurons, astrocytes, and oligodendrocytes. One possibility is that it is many of the same genes across all three cell types. However, if the neurons, astrocytes, and oligodendrocytes all have different genes in these long intergenic regions, that would also be important to know.

Response to Reviewer #3: We thank you for this comment. We are glad that we were able to address the comment. We now tested whether glial-cell-specific genes or neural genes have long intergenic DNA regions. We found that 31.4% of the genes highly induced in each cell type (glutamatergic neuron, astrocytes, and oligodendrocytes) were common in all three cell types (**Figs. 2c, d**). The genes highly induced in all three cell types had significantly long intergenic DNA lengths compared to non-neural genes and were enriched with neural gene ontologies (**Fig. 2e**). The glial-cell-specific genes also had significantly longer intergenic DNA lengths than non-neural genes and had glial-cell-specific gene annotations (**Figs. 2e, f**). Together, we demonstrated that both neural genes and glial-cell-specific genes were enriched around long intergenic regions in the mammalian central nervous system.

REVIEWERS' COMMENTS

Reviewer #1 (Remarks to the Author):

The authors have done an outstanding job of responding to my previous concerns, and I now feel that the resulting manuscript is suitable for publication in Nature Communications.